# Electrochemical grooving of tube inner walls with emphasis on feed strategy and multi-pass effects on material removal and groove geometry

**Semih Ekrem Anil[1], Hasan Demirtas**[ID][2]*

**1** Samsun Yurt Savunma Sanayi, Samsun, Turkiye, **2** Department of Mechanical Engineering, Faculty of Engineering and Natural Sciences, Samsun University, Samsun, Turkiye

* hasan.demirtas@samsun.edu.tr

## Abstract

Electrochemical (EC) grooving minimises tool wear and residual stress when machining hard-to-cut tube materials. This study examines how the number of passes and tool feed direction affect material removal rate (MRR) and removed area (RA) in Stellite 21 tubes. Two feed strategies were tested: Unidirectional Electrolyte Flow (UEF), where the tool moves entirely opposite to the electrolyte flow; and Hybrid Electrolyte Flow (HEF), where the tool first moves against and then with the flow direction. Results showed that the highest MRR values (26.67 mg/s for HEF, 24.8 mg/s for UEF) were observed with two passes, but dropped significantly at four and six passes due to extended machining time. RA increased along the tool path under UEF, reaching up to 327% at the tool exit. Flow simulations revealed that low velocity and conductivity at the tool entry caused under-machining, whereas turbulence at the exit enhanced material removal. These findings offer valuable guidance for optimising EC grooving processes in aerospace and biomedical applications.

## Introduction

Stellite alloys are widely used in applications requiring high resistance to wear and corrosion, such as nuclear power plants, aero engines, biomedical implants, and engine valves. These properties make Stellite alloys attractive candidates for coating materials in cutting tools. However, the same characteristics that provide excellent performance also classify these alloys as difficult-to-machine materials. Conventional machining of Stellite alloys often results in tool wear, residual stress formation, and increased machining time, limiting their manufacturability through traditional methods [1–4].

Given these machining challenges, the difficulties become even more pronounced when internal grooves are to be machined in cylindrical Stellite 21 parts. The high hardness and toughness of Stellite alloys lead to rapid tool wear and elevated cutting forces. Internal surfaces further restrict tool accessibility, heat dissipation, and

**Data availability statement:** All relevant data supporting the findings of this study, are publicly available at: https://doi.org/10.5281/zenodo.15621758.

**Funding:** This study was funded by Savunma Sanayi Baskanligi, Presidency of the Republic of Turkiye (grant number 20SC017). The funders had no role in study design, data collection and analysis, decision to publish, or preparation of the manuscript.

**Competing interests:** The authors have declared that no competing interests exist.

chip evacuation. These limitations often cause dimensional inaccuracies, degraded surface integrity, and thermal damage. Additionally, residual stresses induced during machining can compromise the mechanical performance and durability of components operating under high-temperature or cyclic loading conditions. Electrochemical machining (ECM), a non-contact material removal process, overcomes many of these issues by eliminating tool wear and residual stresses, and is well-suited for producing intricate features in hard-to-machine internal geometries.

Among Stellite alloys, Stellite 21 is predominantly used in casting and hardfacing applications to resist combined corrosive and mechanical attack, particularly under high-temperature conditions—for instance, in valve seats of nuclear power plants and automotive engines. The machining of cylindrical inner-wall geometries is critical for aerospace, oil drilling, and medical systems applications, where high precision and reliability are essential. Given the difficulty of machining these features using traditional approaches, ECM stands out as a promising solution. Its key benefits include high surface quality, absence of thermal damage, and the elimination of mechanical tool wear [5,6]. However, one of the main limitations of ECM is the complex interaction between process parameters and the resulting material removal behaviour, which remains insufficiently understood [7].

Several ECM-based methods have been developed for grooving operations, including Jet-ECM, through-mask ECM, and electrochemical milling (EC milling). These operations can be broadly categorised into two groups: (i) grooving on flat surfaces and (ii) grooving along the inner walls of cylindrical components.

In flat surface grooving, Xu et al. [8] employed shaped sheet tools to machine microgrooves and validated their electric field simulations through experiments. While their study successfully demonstrated the effect of sheet cathode array offsets on current density uniformity, the influence of electrolyte flow rate and temperature distribution was not considered, which may affect groove quality in practice. Liu et al. [9] introduced a mathematical model for Jet-ECM and compared its predictions with finite element method (FEM) simulations and experiments. Despite accurately capturing current density distribution, the model neglected variations in voltage, electrolyte conductivity, and flow rate, limiting its predictive capability under different machining conditions. Zhao et al. [10] designed an L-shaped tool and used FEM to identify the optimal taper angle. Although the approach improved groove quality, the study did not evaluate the effect of dynamic flow or temperature changes, leaving the underlying mechanisms of process stability unaddressed. Similarly, Wang et al. [11] demonstrated that thinner tool sidewalls enhance electrolyte distribution. However, their findings were limited to a fixed geometry, and the influence of varying process parameters on dimensional accuracy was not explored. Additional studies proposed airflow-assisted flushing [12] and modified insulation or tool geometry [13,14], yet these solutions may not translate directly to inner-wall grooving due to different boundary conditions and flow constraints.

ECM of internal cylindrical surfaces poses unique challenges due to restricted accessibility and complex flow dynamics. Chang et al. [15] simulated the electric field for herringbone grooves inside fluid dynamic bearings. However, the study omitted

fluid characteristics, which are crucial for accurately predicting groove formation. Wang et al. [16] employed tool vibration to improve dimensional accuracy of helical grooves. Despite promising results, their analysis did not fully examine regions farthest from the electrolyte inlet, where groove depth deviations were highest. Huang et al. [17] showed that controlling back pressure improves slotting efficiency, but optimal velocity conditions for complete material removal were not fully established. In a separate study, the effect of the electrolyte inclination angle on helix grooving was examined, demonstrating that increased inclination angles resulted in more uniform flow field distributions [18]. As the angle increases, the electrolyte flow gains a rotational component rather than remaining purely linear, which balances the velocity distribution within the machining gap, reduces low-velocity regions, and facilitates the removal of by-products. Gas bubble formation in the gap domain was also modelled for spiral grooves with two [19] and six working teeth [20]. These studies highlighted the benefits of electrolyte inclination and increased cathode teeth for uniform flow, yet the effect of tool feed rate and number of passes on turbulence and flow asymmetry remained uninvestigated. The three-sided tool structure was analysed for straight inner-wall grooving [21], and the results demonstrated that increasing the number of fluid slots improved dimensional accuracy by enhancing heat dissipation and the removal of dissolved material. Process parameters were optimised, with the highest feed rate yielding the most favourable outcomes. However, the study did not perform analyses across any other feed rate values, and neither fluid flow nor temperature distribution analyses were conducted. As a result, the underlying mechanisms causing suboptimal performance at lower feed rates remain uninvestigated. Furthermore, Zhou et al. [22] investigated multiple electrolyte supply units, showing improved flow distribution, though low flow velocities still caused material accumulation and sparking. Tang et al. [23] employed a BP neural network optimised with Particle Swarm Optimisation (PSO) for cathode design. While effective for tool shape prediction, machining parameters such as electrolyte flow rate and tool feed rate were excluded, limiting the model's applicability.

Although previous studies have significantly advanced ECM for internal features, the effects of tool feed direction and number of passes—two parameters that directly influence electrolyte flow dynamics and inter-electrode gap behaviour—have rarely been examined in detail. Addressing this knowledge gap, the present study defines two tool feed strategies, Unidirectional Electrolyte Flow (UEF) and Hybrid Electrolyte Flow (HEF), and systematically investigates their effects on groove formation through both experiments and numerical simulations. To better understand the underlying mechanisms, flow field simulations incorporating a dynamic mesh approach, rarely used in ECM literature, were performed to realistically capture the influence of tool motion on fluid behaviour. Experimental results reveal that although the total amount of dissolved material increases with the number of passes, the material removal rate (MRR) declines after a certain point due to increased electrical resistance in the inter-electrode gap. Moreover, both UEF and HEF strategies exhibit lower electrolyte velocity at the tool entrance region compared to other areas, leading to reduced material removal in that region and resulting in geometric nonuniformity. The feed strategy significantly affects turbulence intensity and electrolyte distribution, thereby influencing groove shape, dimensional accuracy, and overall machining performance. This study contributes to the ECM literature by integrating dynamic-mesh-based simulation with inner-wall grooving experiments and revealing the role of feed strategy in controlling flow asymmetry and process efficiency.

## Materials and methods

Stellite 21 was chosen as the workpiece due to its superior oxidation and high strength at high temperatures. Additionally, Stellite 21 is used in various industries, including the automotive and aerospace industries. The dimensions of the workpiece were 50 mm in length and 13 mm and 20 mm in inner and outer diameters, respectively. The chemical composition and mechanical properties of Stellite 21 are shown in Table 1.

Experimental investigations are carried out using a desktop-sized ECM machine. Detailed specifications of this setup are available in [24], and the flowchart is illustrated in Fig 1. The machine comprises three primary components: (i) the base, (ii) the DC power supply, and (iii) the electrolyte control units. The base is constructed from anodized aluminum profiles to resist oxidation and corrosion. Additionally, the setup is enclosed with plexiglass plates to facilitate the transfer

**Table 1. Chemical composition and the mechanical properties of Stellite 21.**

| Atomic weight (%) | | | | | |
| --- | --- | --- | --- | --- | --- |
| Element | Co | Cr | Ni | Fe | Mo |
| Stellite® 21 | Balance | 25–27 | 2.2–2.5 | 1.3–1.5 | 5.5–6.0 |
| Mechanical Properties | | | | | |
| Hardness (HRB) | | Tensile Strength(MPa) | Modulus of Elasticity (GPa) | | Thermal Conductivity (W/mK) |
| 103 | | 724 | 248 | | 14.7 |

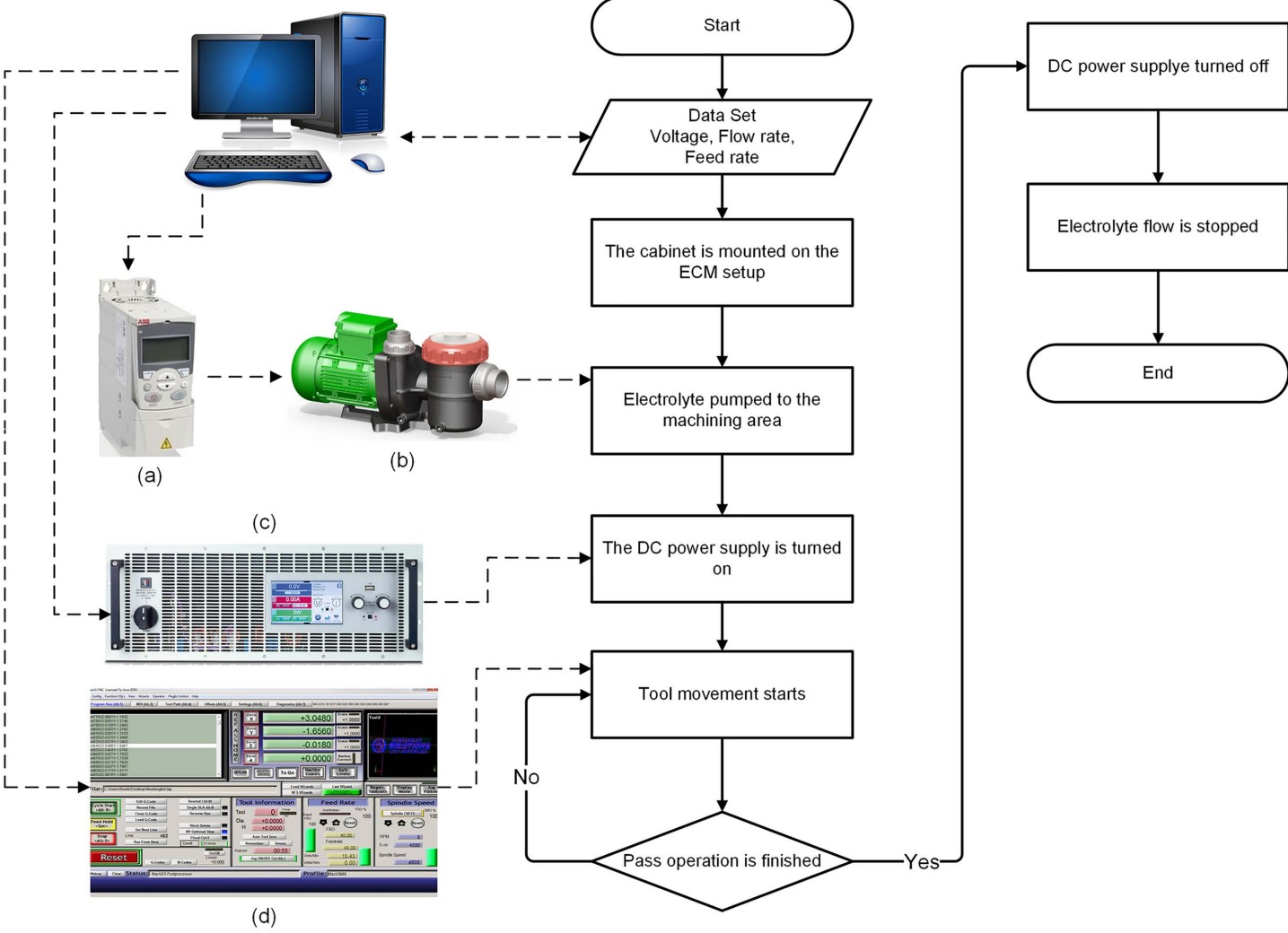

**Fig 1. Flow chart of ECM setup.**

of used electrolyte to the waste tank. As shown in Fig 1, after inputting the data, a computer regulates the frequency inverter (a), DC power supply (c), and Mach3® (d) software. The electrolyte is circulated through the machining area using a 3-phase pump (Nozbart, Turkey), managed by a frequency inverter (ABB-ACS310, Zurich, Switzerland). Subsequently, a DC power supply (c) supplying 60 V and 30 kW power (EA-PSI 10060-1000, Germany) is activated. The motion system

is driven by step motors and managed via the Mach3 (Newfangled Solutions, USA) software, which controls step or ser-vomotor motion by interpreting G-codes.

A closed cabin system with isolating apparatuses was designed for the experiments. A brass chamber was intended to house a cable connection to the workpiece and was placed inside the cabin system. The workpiece was inserted into the brass chamber and then placed inside the cabin's lower case (Fig 2a). The tool front was positioned at X = 0, where the electrolyte leaves the workpiece (Fig 2b). The tool, made of brass, had an octagonal structure, and specific lengths of insulation material were placed in corners where machining was not desired. A conical apparatus was placed at the front of the tool to make the electrolyte transition more uniform (Fig 2c).

EC grooving is a complex machining process in which various parameters directly affect system performance. As dis-cussed above, ensuring a well-distributed electrolyte is essential for achieving high material dissolution and dimensional accuracy. Therefore, the number of passes and the tool feed direction were chosen to investigate their effect on mate-rial dissolution behavior and geometrical parameters. Under high humidity, brass tends to be damaged via passive film formation in $H^+$ and $NO_3^-$ solutions [25], which could negatively affect machining accuracy and tool durability. Therefore, experiments were conducted under controlled ambient conditions, maintaining the temperature at approximately 25°C and relative humidity below 50% to prevent corrosion and ensure consistent machining performance. The parameters used in this study were selected based on the preliminary experiments. The voltage of 20 V and electrolyte flow rate of 4.0 l/min were found to provide a good balance between efficient material removal and surface quality without excessive gas forma-tion or instability, consistent with previous ECM studies [26] and shown in Table 2.

An investigation of the effect of electrolyte transfer direction on electrochemical dissolution and groove geometry was conducted using two different tool feed directions. These methods were as follows: (i) the tool was fed at 45 mm/min in the opposite direction of the electrolyte flow (Fig 2e), and (ii) the tool was fed at 45 mm/min in the opposite direction of the electrolyte flow, and upon completion of the path, it returned to the starting point at the same feed rate (Fig 2d). Therefore, machining was carried out in the same and opposite directions of the electrolyte flow in the second case. In the following sections, these movement methods are called unidirectional electrolyte flow (UEF) and hybrid electrolyte flow (HEF). For both UEF and HEF conditions, the tool is first positioned at the X = 0 point and moves through the X = 50 mm at a 45 mm/min feed rate. After the tool reaches X = 50, it returns to the operation starting point (X = 0) at different feed rates. These feed rates are 600 mm/min (Fig 2e) and 45 mm/min (Fig 2d) for UEF and HEF conditions, respectively. Hence, the UEF condition completes a 1-pass operation while the HEF condition completes a 2-pass operation. In this state (X = 0), it is checked whether the process is repeated or not. If it is not repeated, the tool moves to the X = 50 point for UEF at a 45 mm/min feed rate, thus repeating the process once more. As a result, after the 2-pass operation is completed for both UEF and HEF conditions, this process is repeated for 4- and 6-pass operations. A flow chart of the EC grooving process is shown in Fig 3.

The dependent variables in the experiments were the amount of dissolved material and the Removed Area (RA) for dimensional accuracy. The dissolved material and material removal rate (MRR) were calculated using the following equations:

$$\Delta m_j = m_b - m_a \qquad (1)$$

and

$$MRR = \frac{\Delta m_j}{t_t} \qquad (2)$$

where $t_t$ is the total machining time, j is the number of passes, $\Delta m_j$ is the dissolved material, and $m_b$ and $m_a$ are the mass of the workpiece before and after the experiments.

 

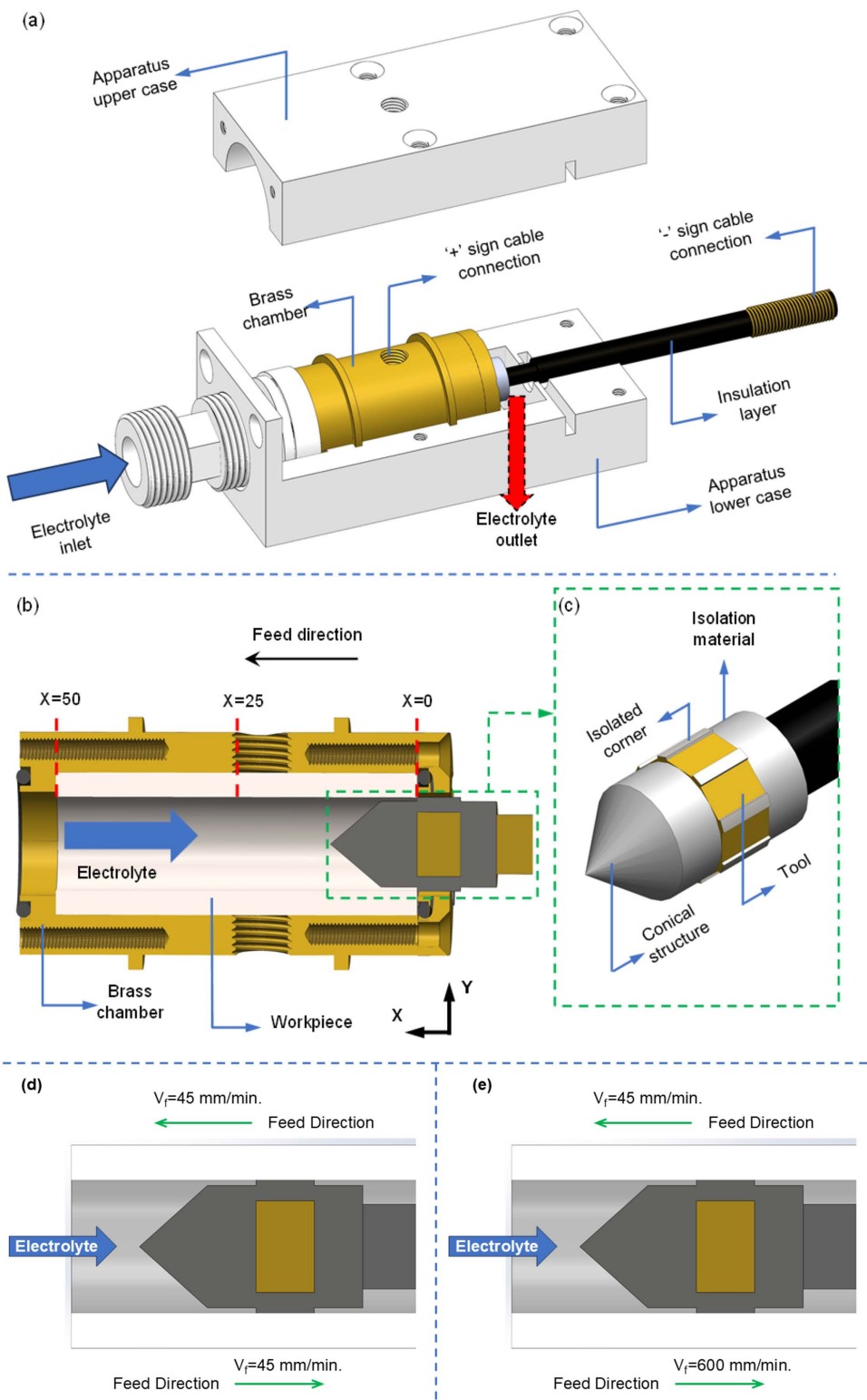

**Fig 2. Schematic illustration of the cabin system (a); cross-sectional view of the machining area (b); detailed view of the tool (c); HEF strategy (d); and UEF strategy (e).**

**Table 2. Constant EC grooving parameters.**

| Electrolyte Type | Electrolyte Conductivity (mS/cm) | Electrolyte Temperature (⁰C) | Voltage (V) | Flow Rate (l/min.) | Ambient Temperature (⁰C) | Relative Humidity (%) |
|---|---|---|---|---|---|---|
| $NaNO_3$ | 105 | 25 | 20 | 4.0 | 25 | < 50 |

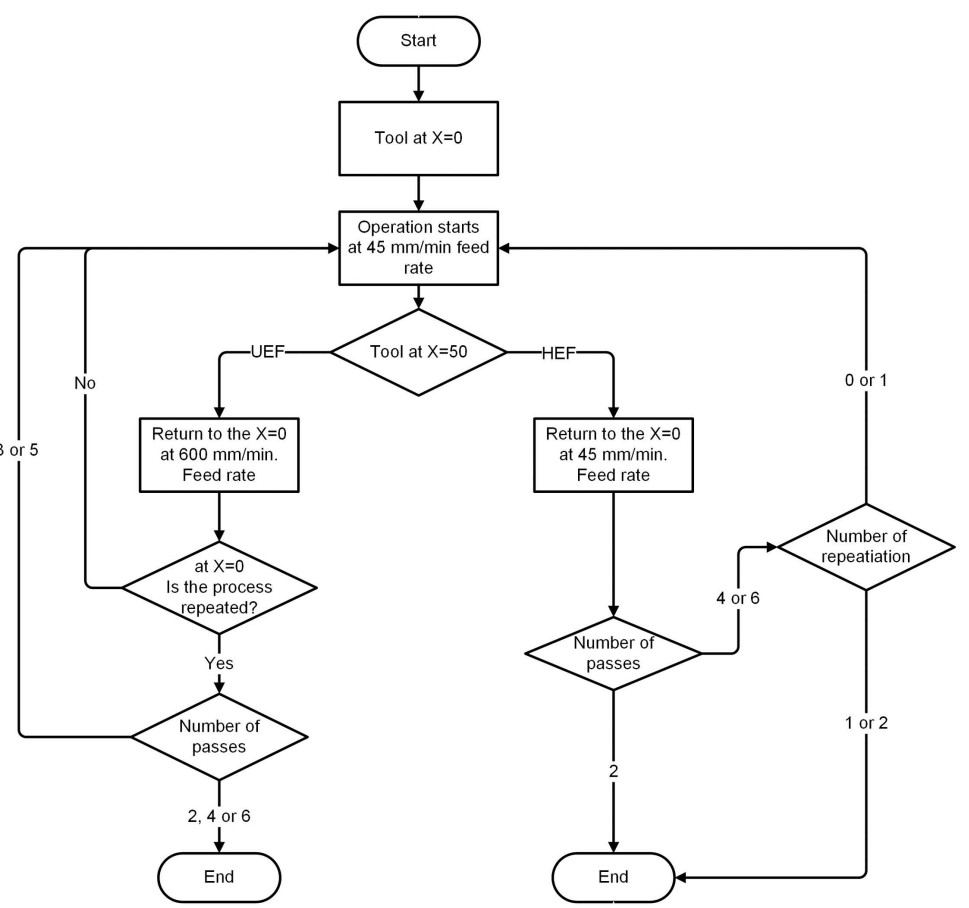

**Fig 3. Flow chart of EC grooving operation.**

The EC-grooved workpieces were cut using a wire electrical discharge machine (W-EDM). A 1-mm-thick section of the workpiece was taken, and its geometries were measured using a video measuring system (Nikon NEXIV VMA-2520). Since the measurement values obtained were superficial and not volumetric, the machined amount was determined and compared in terms of area. In addition, to determine how the machining changed across the entire axial direction, separate measurements were taken where the tool entered, in the middle of the workpiece, and where the tool exited, and compared—the section where the grooves were machined. Cross-sectional images of these sections are shown in Fig 4.

The volume of material removed from the workpiece was determined using the following formula:

$$\Delta V_j = \frac{\Delta m_j}{\rho_w} \tag{3}$$

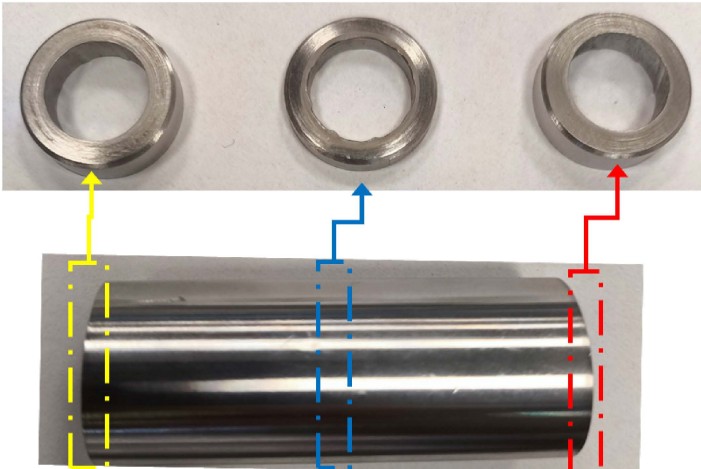

**Fig 4. Image of the workpiece and cross-sections of the part.**

where $\Delta V$ is the change in volume of removed material, and $\rho_w$ is the density of the workpiece. The following equation was used to calculate Eq. 3 as a change in area:

$$A_j = \frac{\Delta V_j}{l}$$

(4)

where A is the calculated RA value and l is the length of the workpiece (50 mm). The experimental conditions, including the input and output parameters, are shown in Table 3.

## Numerical simulation of electrolyte flow

### Modelling of the fluid domain

Fig 5 shows the fluid domain of the EC grooving system derived from its CAD model. The yellow and red zones indicate the electrolyte inlet and outlet, respectively. The purple, blue, and gray domains represent the walls of the cathode, the anode, and the apparatus. The machining gap between the two electrodes is 0.7 mm. The Cross-sectional plane A is defined within the fluid model to evaluate the distribution of electrolyte velocity. In EC grooving, a turbulent electrolyte flow is typically necessary to facilitate the rapid removal of heat and electrolysis by-products [27]. Based on the characteristics of the flow field, the realizable $k-\varepsilon$ turbulence model is adopted to simulate the electrolyte flow behavior [21]. The governing equations for turbulent kinetic energy and its dissipation rate under irregular flow conditions are given as follows:

$$\frac{\partial(\rho k)}{\partial t} + \frac{\partial(\rho k u_i)}{\partial x_i} = \frac{\partial}{\partial x_j}\left[(\mu + \frac{\mu_t}{\sigma_k})\frac{\partial k}{\partial x_j}\right] + G_k - \rho\varepsilon$$

(5)

$$\frac{\partial(\rho\varepsilon)}{\partial t} + \frac{\partial(\rho\varepsilon u_i)}{\partial x_i} = \frac{\partial}{\partial x_j}\left[(\mu + \frac{u_t}{\sigma_\varepsilon})\frac{\partial\varepsilon}{\partial x_j}\right] + \rho\varepsilon C_1 E - \rho C_2 \frac{\varepsilon^2}{k + \sqrt{v\varepsilon}}$$

(6)

where v is the kinematic viscosity, $\varepsilon$ is the dissipation rate, k is the turbulence kinetic energy, $\rho$ is the density of the electrolyte, $x_i$ and $x_j$ are the spatial coordinates, and $u_i$ and $u_j$ are the ith and jth components of the velocity vector, respectively. The constant values in Eqs. 5 and 6 are presented in Table 4.

**Table 3. Results of the experimental conditions for Stellite 21.**

| Exp. Cond. | Feed Movement Type | Number of Passes | Δm (gr) | A (mm²) |
|---|---|---|---|---|
| 1 | UEF | 2 | 3.68 | 8.86 |
| 2 | | 4 | 5.88 | 14.16 |
| 3 | | 6 | 7.99 | 19.22 |
| 4 | HEF | 2 | 3.47 | 8.36 |
| 5 | | 4 | 5.77 | 13.88 |
| 6 | | 6 | 7.74 | 18.62 |

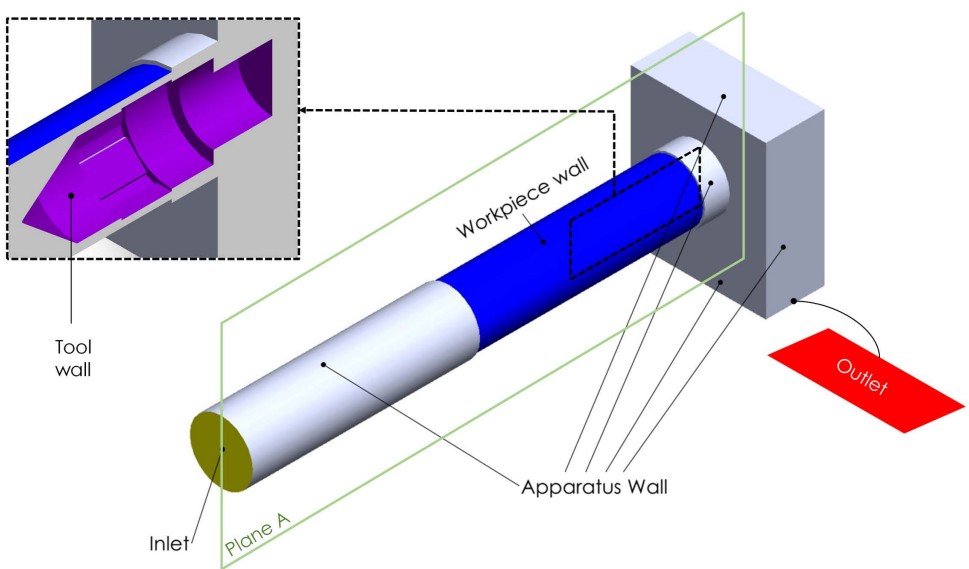

**Fig 5. Fluid model of EC grooving.**

The turbulent viscosity $\mu_t$ and average velocity gradient $G_k$ are obtained as follows:

$$\mu_t = \rho C_\mu \frac{k^2}{\varepsilon}$$

(7)

$$G_k = \mu \left( \frac{\partial u_i}{\partial x_i} + \frac{\partial u_j}{\partial x_j} \right) \frac{\partial u_i}{\partial x_j}$$

(8)

The simulations disregarded the effects of electrolysis products, temperature fluctuations, and gas bubbles on the flow field to streamline the computational process. Additionally, the electrolyte was treated as an incompressible and continuous fluid [21]. The movement of the electrolyte is dictated by the conservation laws of mass and momentum, represented by the following equations:

$$\frac{\partial \rho}{\partial t} + \frac{\partial (\rho u_i)}{\partial x_i} = m_s$$

(9)

**Table 4. Constant values for the $k - \varepsilon$ model.**

| $\sigma_\varepsilon$ | $\kappa$ | $C_2$ | $C_1$ |
|---|---|---|---|
| 1.2 | 1 | 1.9 | $\max\left\{0.43, \frac{\eta}{(\eta+5)}\right\}$ |

$$\frac{\partial(\rho u_i)}{\partial x_i} + \frac{\partial(\rho u_i u_j)}{\partial x_j} = \frac{\partial \tau_{ij}}{\partial x_j} - \frac{\partial P}{\partial x_i} + F_i + \rho g_i$$

(10)

where $m_s$ is the mass of the dispersed secondary phase added to the continuous phase, $\tau_{ij}$ is the stress tensor, P is the pressure, $F_i$ is volumetric forces, $\rho g_i$ is the gravity volume force.

## Boundary conditions and meshing

The boundary and initial conditions of the flow field model constrain and direct the electrolyte flow, serving as essential parameters for completing the flow field calculations. Typically, the boundary conditions include the electrolyte inlet and outlet, wall surfaces, and internal interfaces. The electrolyte, carrying electrolysis products, exits the machining zone through the outlet at the lower shaft, which is directly connected to the ambient environment. Therefore, the electrolyte outlet is defined as a pressure outlet boundary condition. All surfaces other than the inlet and outlet are treated as wall boundaries, forming the flow domain. The electrolyte is considered an incompressible fluid, and the corresponding pressure relationship is defined as follows:

$$P_{outlet} = P_{inlet} - \frac{\rho u^2}{2}$$

(11)

where $\frac{\rho u^2}{2}$ is the dynamic pressure, $P_{outlet}$ and $P_{inlet}$ are the outlet and inlet pressure respectively.

Mesh generation is a critical step in the numerical simulation of the flow field, as mesh quality directly influences both the accuracy and efficiency of the calculations. Given the complex and variable nature of the flow field, tetrahedral elements are used for meshing. The minimum and maximum element sizes are set to 0.08 mm and 1 mm, respectively. Curvature and proximity capturing techniques ensure smooth transitions within the machining area. Boundary layers are defined at the inlet and outlet to enhance simulation precision. Mesh refinement is applied in critical regions such as the machining zone and cathode walls. Fig 6 shows the meshed model, which consists of approximately 4,583,431 elements with a maximum skewness value of 0.85.

The tool return feed rate defines the method of tool movement (HEF or UEF), as discussed earlier. Accordingly, the dynamic mesh method is employed to analyze the electrolyte velocity distribution within the machining zone. This approach enables the simulation of fluid flow in domains with changing geometries. In the dynamic mesh method, the computational mesh is updated at each time step of the transient simulation by adjusting the positions of mesh nodes to follow the deformation of moving boundaries [28]. To prevent mesh distortion and ensure accurate results, optimal mesh sizes and time step values are carefully selected, as summarized in Table 5.

The most significant difference in RA between UEF and HEF occurs at the tool entry point (X = 0). Therefore, fluid analysis is conducted at the tool's entry point to the workpiece. This analysis begins when the rear end of the tool aligns with the workpiece entrance and continues until the back of the tool aligns with the entry point. Due to varying feed rates, the tool requires different durations to cover the same distance: 0.6 seconds for UEF and 8 seconds for HEF. As shown in Table 5, these durations correspond to 60 time steps for UEF and 800 time steps for HEF. To comprehensively assess the influence of electrolyte velocity, it is also essential to examine the tool's motion in the opposite direction. Once the front of the tool reaches the workpiece entrance, it continues moving through the workpiece at a constant feed rate of 0.0075 m/s

 

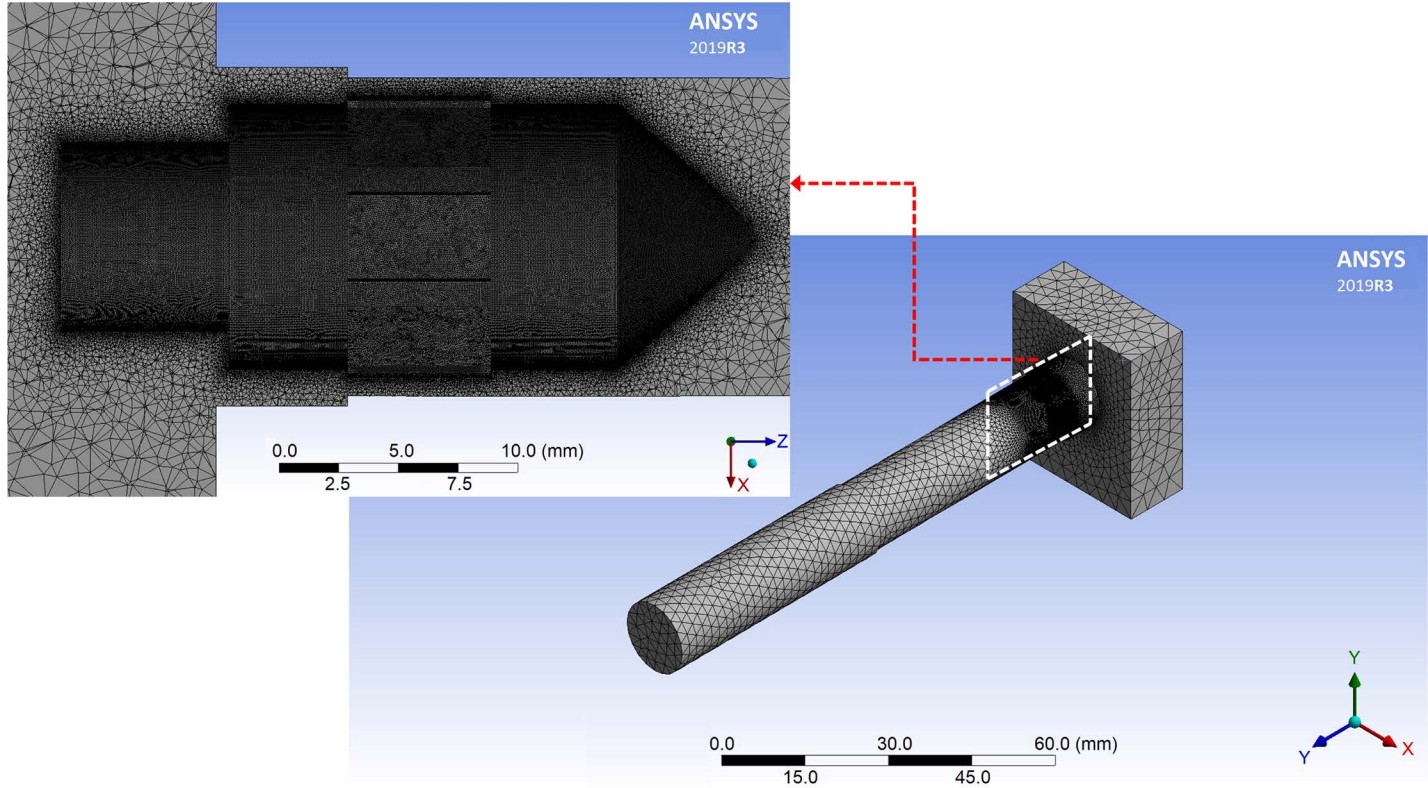

**Fig 6. Meshed fluid domain (right) and tool wall (left).**

**Table 5. FEM analysis parameters.**

| Feed Direction | Turn Feed rate (m/s) | Number of time steps | Max. and Min. element size (mm) | Skewness | Electrolyte Inlet Velocity (m/s) | Time Step (s) |
|---|---|---|---|---|---|---|
| UEF | 0.1 | 80 | 1-0.08 | Max 0.85 | 0.25 | 0.01 |
| HEF | 0.0075 | 820 | | | | |

under both UEF and HEF conditions. Consequently, the simulation is extended by an additional 20 time steps, leading to a total of 80 time steps for UEF and 820 for the HEF condition. The tool position for the mentioned time steps is shown in Fig 7.

## Results and discussions

### Analysis of material dissolution

The results and variations in current over time are shown in Fig 8. The MRR (mg/s) and removed material (g) are indicated with arrows for each experimental condition. Evaluating each graph individually, it can be seen that the current decreases by similar amounts in each pass. For example, the current values decreased from 150 A to 120 A in the first two passes. Current decreased to 100 A in the third pass, 90 A in the fourth pass, 85 A in the fifth pass, and 80 A in the sixth pass. Accordingly, with a 30 A change, in the UEF condition, the removed material amount is 3.68 g, while in the HEF condition, this amount is 3.47 g. For four passes with a total change of 60 A compared to the initial condition, the values

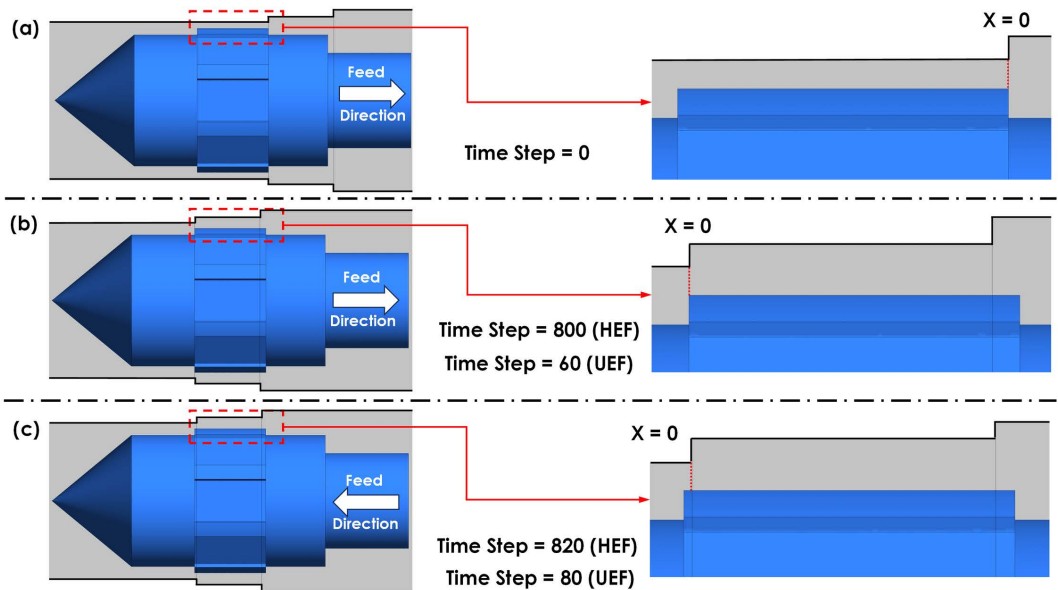

**Fig 7. Tool positions at different time steps: (a) the back of the tool at X=0, (b) the front of the tool at X=0, and (c) after 20 time steps, when the front of the tool passes X=0.**

were 5.88 g and 5.77 g, respectively. In the six passes where the most machining occurred, the values were 7.99 and 7.74 g, and the total current variation was around 80 A.

It can be observed from Fig 8 that although the cumulative amount of dissolved material increases, the amount removed in each additional pass decreases. This is because the change in the amount of dissolved material does not occur linearly with increasing machining time and number of passes. Based on the initial current value of 150 A, the current reduction is approximately 20% after two passes, 40% after four passes, and 48% after six passes. This reduction can be attributed to the tool movement strategy. As described in the Materials and methods section, the tool moves along the X-direction (Fig 2b) relative to a workpiece with cylindrical geometry, while no movement occurs in the Y-direction (Fig 2b). Consequently, with each pass, the distance between the tool and the workpiece increases. This increased gap results in higher electrical resistance in the machining gap. The electrical resistance R in an electrolyte circuit can be calculated using the following equation:

$$R = \frac{L}{k_c . A}$$

(12)

where L is the distance between the electrodes, $k_c$ is the electrolyte conductivity, and A is the cross-sectional area between the tool and the workpiece. As the distance L increases with each pass, the resistance increases, which limits the current flow and thus reduces the amount of material dissolved in subsequent passes.

As seen in Fig 8, the material removal rate (MRR) decreases with an increasing number of pass operations. According to Faraday's laws, the MRR can be calculated using the following equation:

$$MRR = \frac{J . k_v}{F . \rho_w}$$

(13)

where J is the current density, $k_v$ is the electrochemical machinability of the workpiece, *F* is the Faraday constant, and $\rho_w$ is the density of the workpiece. In this study, all experiments were conducted using the same workpiece material, ensuring

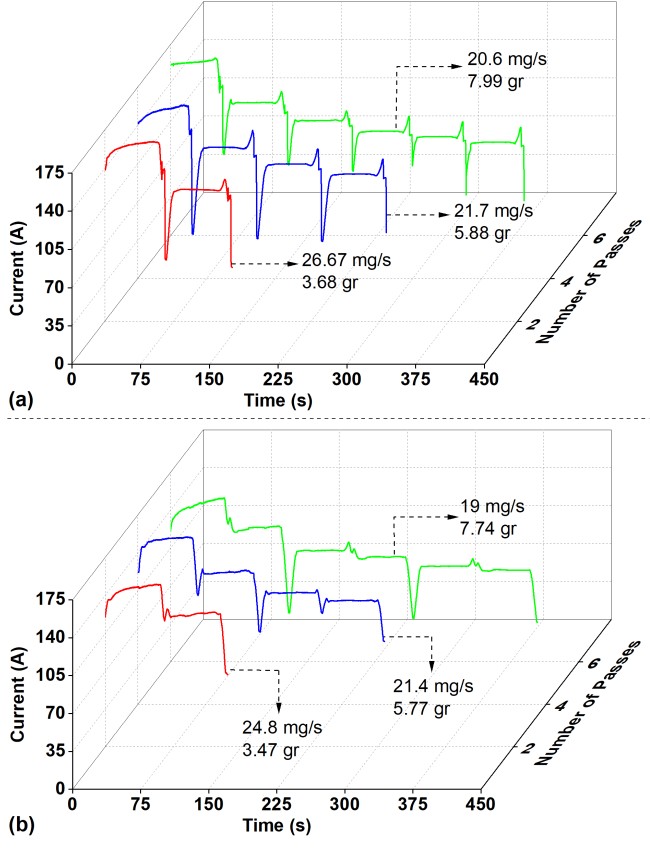

**Fig 8. Current variation over time for (a) the UEF condition and (b) the HEF condition.**

that both $k_v$ and ρ remained constant. Therefore, the MRR is directly influenced by the current density J, which can be determined using Eq. 14:

$$J = \frac{I}{A}$$

(14)

where I is the machining current and A is the mean cross-sectional area of the machined region, derived from RA measurements in the cross-sectional images (see Fig 4), and can be calculated using Eq. 4. As shown in Fig 8, the machining current I decreases; decreases as the number of passes increases. Consequently, the current density J also decreases, leading to a reduction in MRR with each additional pass.

In Fig 9, ΔM represents the difference in dissolved material between two passes and is calculated using the following equation:

$$\Delta M_{i+2} = \Delta m_{i+2} - \Delta m_i$$

(15)

where Δm is the amount of dissolved material, and i is an integer that can be 0, 2, or 4. Although the cumulative amount of dissolved material increased consistently, the highest ΔM values were observed during the first two passes under both tool movement conditions. The differences decreased with each subsequent pass. Compared to the first two passes, ΔM decreased by approximately 40% and 34% in the four-pass condition, and by 43% and 44% in the six-pass condition,

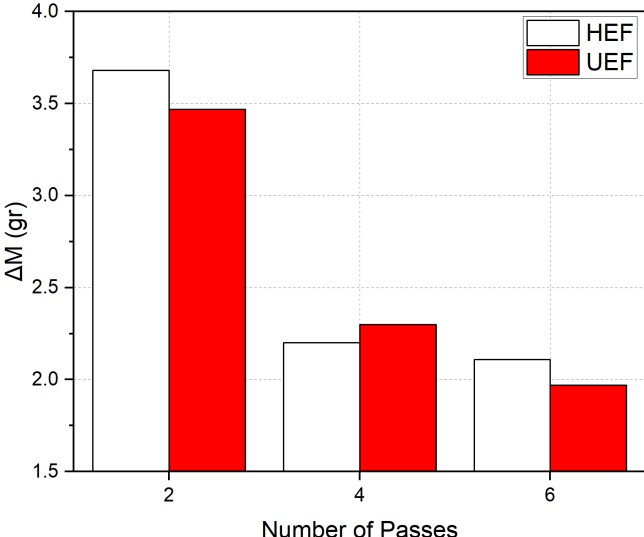

**Fig 9. Change of dissolved material (ΔM) for the UEF and the HEF conditions.**

for the UEF and HEF strategies, respectively. These findings indicate that the material dissolution rate diminishes as the number of passes increases. An analysis of the cross-sectional geometry in the radial direction reveals that the groove formed during the first two passes, when the influence of the insulation material was more pronounced, had a more regular shape. As the number of passes increased, both the groove depth and width expanded. Notably, the effect of the insulation material declined significantly, particularly after six passes, leading to an enlarged inner diameter due to over-machining. Fig 10 illustrates these geometric changes under the UEF condition. The groove geometries were obtained from the central section of the workpiece.

As seen in Fig 9, the UEF condition resulted in the highest amount of dissolved material and the largest machined area. This outcome is primarily attributed to the tool movement direction and the characteristics of the electrolyte flow. In the HEF strategy, the tool moves at a constant speed, regardless of the electrolyte flow direction. In contrast, in the UEF strategy, the tool moves at a constant feed rate in the direction opposite to the electrolyte flow and rapidly returns to its starting position at 600 mm/min in the same direction as the flow. As a result, machining occurs only during the tool's movement opposite to the flow direction. This unidirectional interaction, combined with the conical shape of the tool tip, forces the electrolyte to flow through the gap between the tool and the workpiece. This forced flow creates vortices along the inner

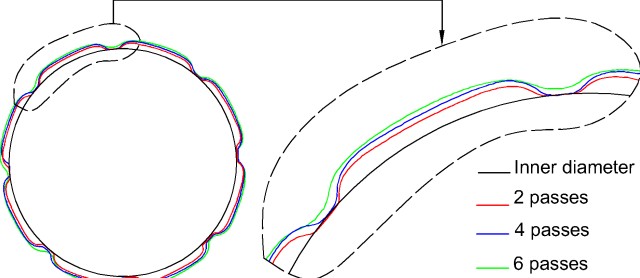

**Fig 10. Changes in groove geometry for different numbers of passes.**

wall of the workpiece due to increased pressure and velocity. Under the HEF condition, half of the tool passes are against the flow direction, while the other half are in the same direction. During the latter, the electrolyte fills the machining gap more easily, which reduces material removal efficiency. A schematic illustration of this process is shown in Fig 11.

## Analysis of the geometrical parameters

Fig 12 presents the RA data for different pass numbers under both HEF and UEF conditions. Hollow symbols indicate the number of passes measured using projection, while the calculated area values (based on Eq. 4) are represented by colored symbols on the graph.

In Fig 12, X = 0 marks the point where the tool enters the workpiece, and X = 50 denotes the tool exit point; these also correspond to the electrolyte exit and entry points, respectively. Both tool feed strategies (UEF and HEF) were evaluated, and the RA values increase with increasing X distance, which aligns with findings reported in the literature [16]. Consequently, the calculated machining area ($A_j$) under any experimental condition does not exactly match the values observed at varying X positions. As discussed in the Analysis of material dissolution section, the amount of dissolved material increases with the number of passes, thereby influencing the RA. However, discrepancies were observed particularly at the tool entry region. The subsequent sections provide a comparative analysis of these differences with respect to the tool feed strategies.

**Hybrid electrolyte flow.** Fig 12a shows that the increase in RA's starting and ending values was around 71% for two passes, 43% for four passes, and 104% for six passes. To better evaluate the data, the average values of $A_j$ for two, four, and six passes are shown in Fig 13.

As shown in Fig 13, since the $A_2$ value for two passes is close to the average, a similar process occurred as shown in Fig 12a. Despite this, $A_4$ and $A_6$ are above the mean of RA for four and six passes. Therefore, it is predicted that the machining will stabilize before the middle region of the workpiece. It can be said that the machining is more stable than the 2-pass machining, since the change of groove geometry is not much for the 4-pass machining. In addition, as seen in Fig 12a, the obtained $A_4$ value gives similar results in the regions close to the exit region of the tool. In a 6-pass operation, the expansion continues toward the middle of the material and remains stable from the middle to the end. According to the obtained data, the material is machined less in the part where the tool enters than in the other parts, and it is machined more in the part where it exits. This can be attributed to the electrolyte's fluid dynamics, which change suddenly, resulting in a velocity difference. Fig 14a shows the electrolyte velocity variation with time step obtained by the FEM analysis at X = 0 where the tool enters the workpiece. As illustrated in Fig 14a, the electrolyte velocity fluctuates up to 800 steps when the tool front is at the X = 0 position. As mentioned in the Materials and methods section, the part where the electrolyte moves away from the designed apparatus is closer to the part where the tool enters the workpiece. Therefore, the electrolyte velocity decreases continuously where the electrolyte pressure is closest to the external environment pressure. J. Liu et al. [29] found that higher electrical conductivity is achieved at higher electrolyte velocities due to rapid transportation of dissolved material. Therefore, minimum electrical conductivity can be achieved at X = 0, rather than at X = 25 and X = 50 points. According to the Faraday and Ohm laws, electrical conductivity directly affects the amount of material dissolution, as shown in Eqs. 12–14, thus, min. RA values obtained at X = 0 for both conditions are shown in Fig 12. Additionally, high RA values are obtained due to the high-pressure difference in the area where the tool exits at X = 50. To prevent this, channels can be added in the inlet and outlet of the electrolyte to balance electrolyte pressure. The highest RA values were obtained in the six-pass operation due to the increase in the material's inner diameter. As previously mentioned, the RA values did not change much at distances of X = 25–50 mm as the distance between the tool and the workpiece increased. Fig 14b shows an enlarged view of the electrolyte velocity distribution after 800 time steps, corresponding to the moment when the tool begins feeding into the workpiece. This snapshot captures the electrolyte flow behavior immediately after the tool returns to the initial entry point. It continues for an additional 20 time steps, providing insight into the dynamic conditions at the onset of machining.

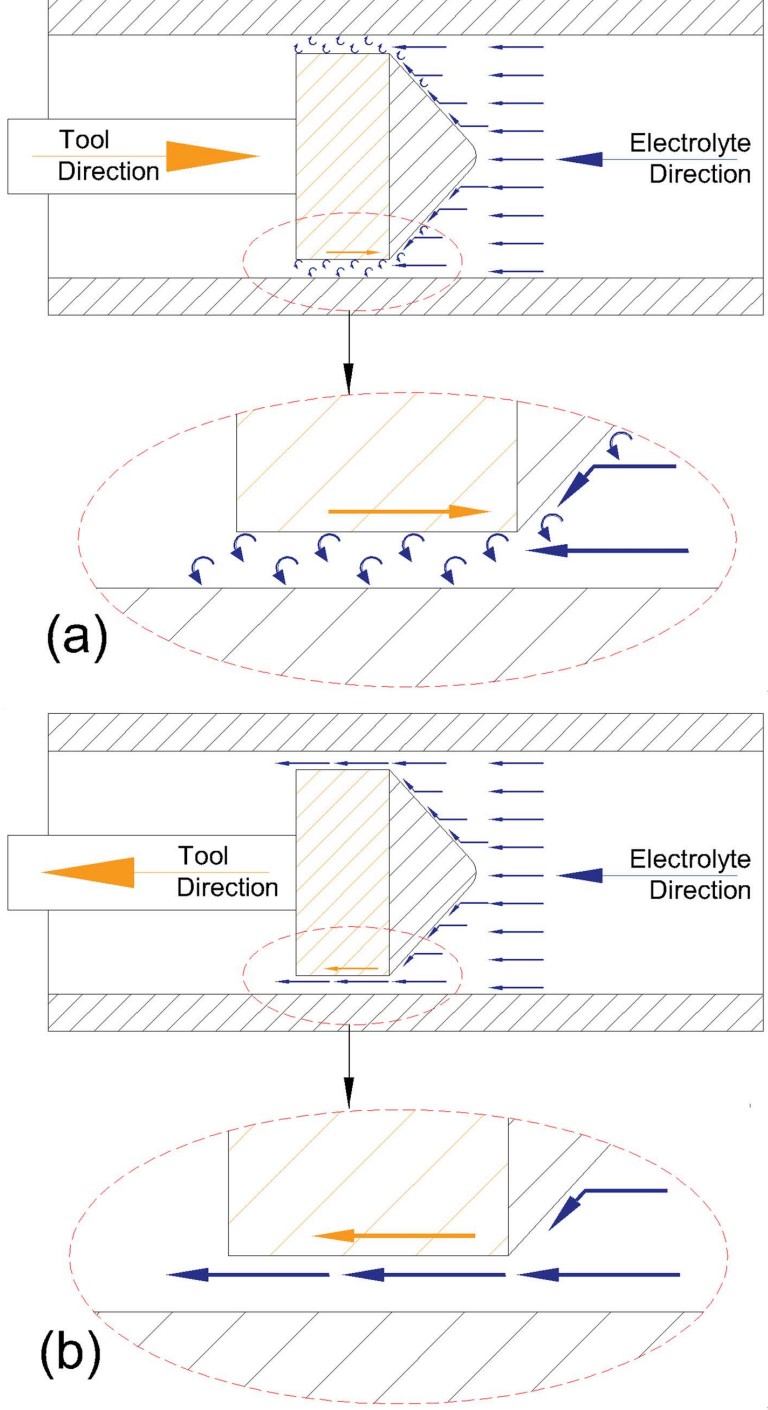

**Fig 11. Schematic of a material dissolution showing when the tool moves (a) in the opposite direction of the electrolyte flow and (b) in the same direction as the electrolyte flow.**

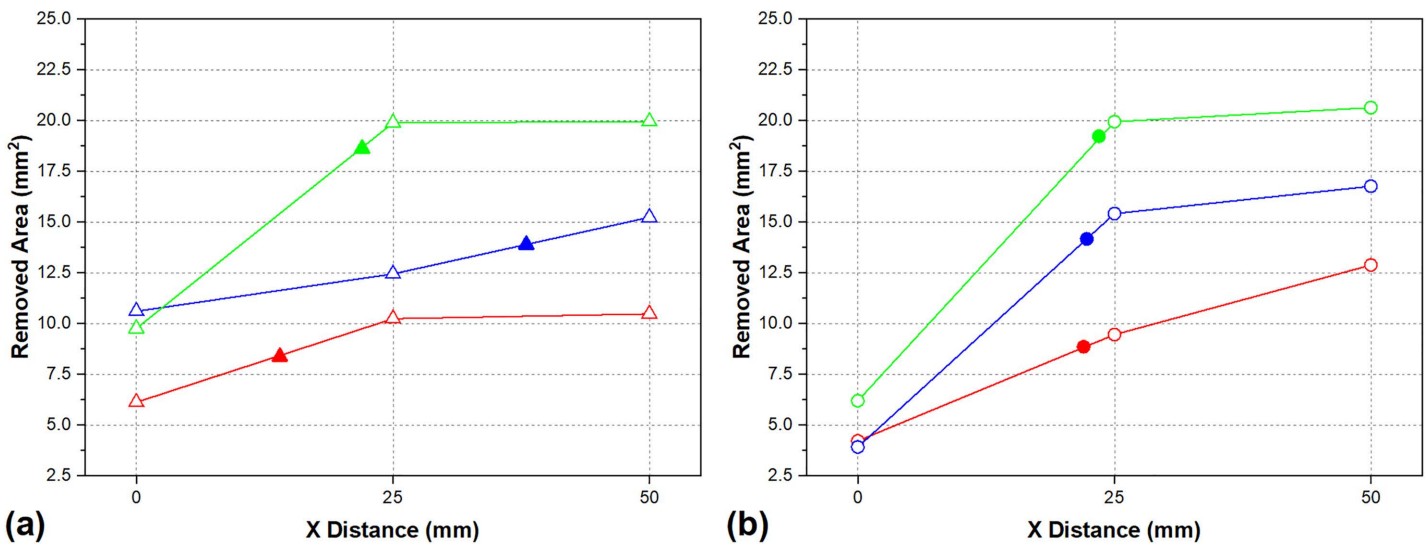

**Fig 12. Variations in RA with X distance for (a) HEF and (b) UEF.** The line color describes the number of passes (red line 2 passes, blue line 4 passes, and green line 6 passes). Shapes filled with color dedicate the calculated RA ($A_j$).

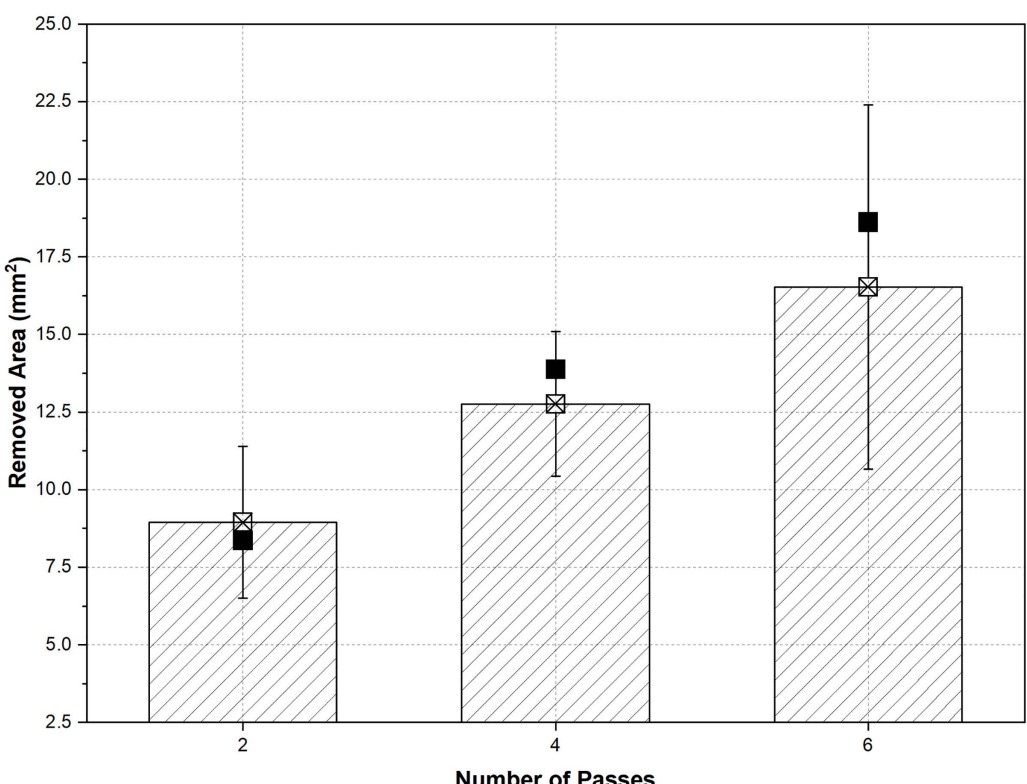

**Fig 13. Mean of the RA (with X sign) and error lines for the different pass operations with $A_j$ (with filled ones).**

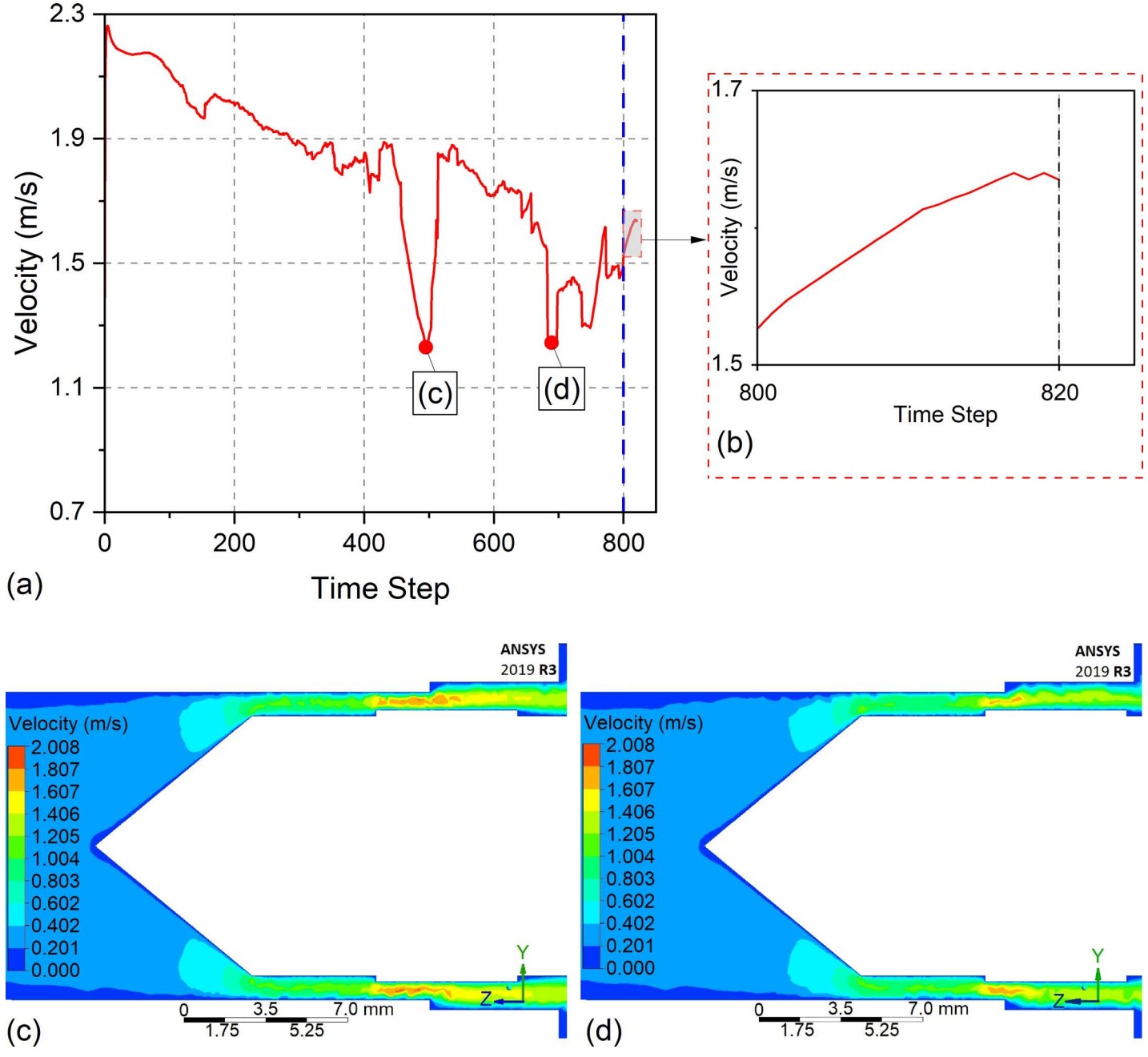

**Fig 14. Electrolyte velocity variation at X = 0 as a function of time steps under the HEF condition: (a) overall velocity variation; (b) enlarged view highlighting the velocity behavior at the onset of tool feeding; (c–d) velocity distributions at specific time steps corresponding to the fluctuations observed in (a).**

Fig 14c and 14d illustrate the electrolyte velocity distribution at specific time steps corresponding to the flow fluctuations identified in Fig 14a. The results reveal that the velocity profile is not yet fully developed at these moments, which may account for the variations in the RA observed in Fig 12a.

**Unidirectional electrolyte flow.** In the analysis of the UEF condition, the increased amounts of RA for 2, 4, and 6 passes were 205%, 327%, and 232%, respectively. In the UEF condition, the tool operates at a specific speed (45 mm/min) in the entrance region (i.e., low electrolyte pressure). The electrolyte velocity variation with time step in the tool entry point at X = 0 for UEF is shown in Fig 15a, and the enlarged velocity variation after 60 time steps is shown in Fig 15b. As shown in Fig 15a, velocity decreases until the tool front achieves X = 0 that similar results have been observed with HEF (Fig 14a) at X = 0 (at time step 60), where the tool exits the workpiece. This result confirms the validity of the proposed approach discussed in the Hybrid electrolyte flow section. However, the reduction rate of the velocity is more uniform and does not fluctuate as observed under the UEF condition. This shows that machining is made uniformly at the X = 0 position. According to the Eulerian velocity field, it can be expressed as follows:

$$\mathbf{V} = u(x,y,z,t)\mathbf{i} + v(x,y,z,t)\mathbf{j} + w(x,y,z,t)\mathbf{k} \tag{16}$$

where **V** is the velocity, u, v, and w are the velocity components along the x, y, and z directions. Thus, the acceleration field is:

$$\mathbf{a} = \frac{d\mathbf{V}}{dt} = \frac{du}{dt}\mathbf{i} + \frac{dv}{dt}\mathbf{j} + \frac{dw}{dt}\mathbf{k} \tag{17}$$

can be written by using the chain rule:

$$\mathbf{a} = \frac{\partial \mathbf{V}}{\partial t} + \left( u\frac{\partial \mathbf{V}}{\partial x} + v\frac{\partial \mathbf{V}}{\partial y} + w\frac{\partial \mathbf{V}}{\partial z} \right) \tag{18}$$

where **a** is the acceleration of the fluid, $\frac{\partial \mathbf{V}}{\partial t}$ is the local acceleration and $u\frac{\partial \mathbf{V}}{\partial x} + v\frac{\partial \mathbf{V}}{\partial y} + w\frac{\partial \mathbf{V}}{\partial z}$ is the convective acceleration. In fluid dynamics, the local acceleration results when the flow is unsteady [30]. After 60 time steps, the electrolyte velocity continues to increase in a similar trend; however, the rate of increase is lower compared to the HEF condition (Fig 14b). This variation in velocity over time reflects the local acceleration of the electrolyte. The results suggest that under the UEF condition, the electrolyte flow is more uniformly distributed than in the HEF condition. Fig 15c and 15d present the electrolyte velocity distribution when the tool is positioned identically to that in Fig 14c and 14d, respectively. These simulation results further validate that the velocity distribution under UEF is more homogeneous, which explains the consistent RA values observed at X = 0 across different pass operations (Fig 12b).

The RA values were higher since the tool movement took longer in the HEF condition. The amount of RA in two passes increased continuously and did not change much beyond a distance of X = 25 mm for four and six passes. Fig 16 shows the mean of the RA with error lines for the different pass operations. The error range is higher than the HEF condition, which is attributed to the RA difference at the X = 0 position for all pass operations. It can be seen that the $A_j$ value, which is close to the average, shows that the workpiece is machined linearly (Fig 12a).

The values of $A_4$ and $A_6$ are higher than the average RA in the 4- and 6-pass operations. As shown in Fig 12a, and given that the RA values measured at 25- and 50-mm distances are close to each other, it is clear that the machining becomes stable before the midpoint (X = 25 mm).

Generally, it is observed that the geometry stabilizes before the middle region of the workpiece for all numbers of passes, except for two. The Analysis of material dissolution section mentioned that the inner diameter of the workpiece, which was exposed to more machining time in the 4- and 6-pass operations, increased.

## Conclusions

This study investigated the effects of electrolyte flow direction—Unidirectional Electrolyte Flow (UEF) and Hybrid Electrolyte Flow (HEF)—and the number of passes on electrochemical machining (ECM) performance for grooving operations

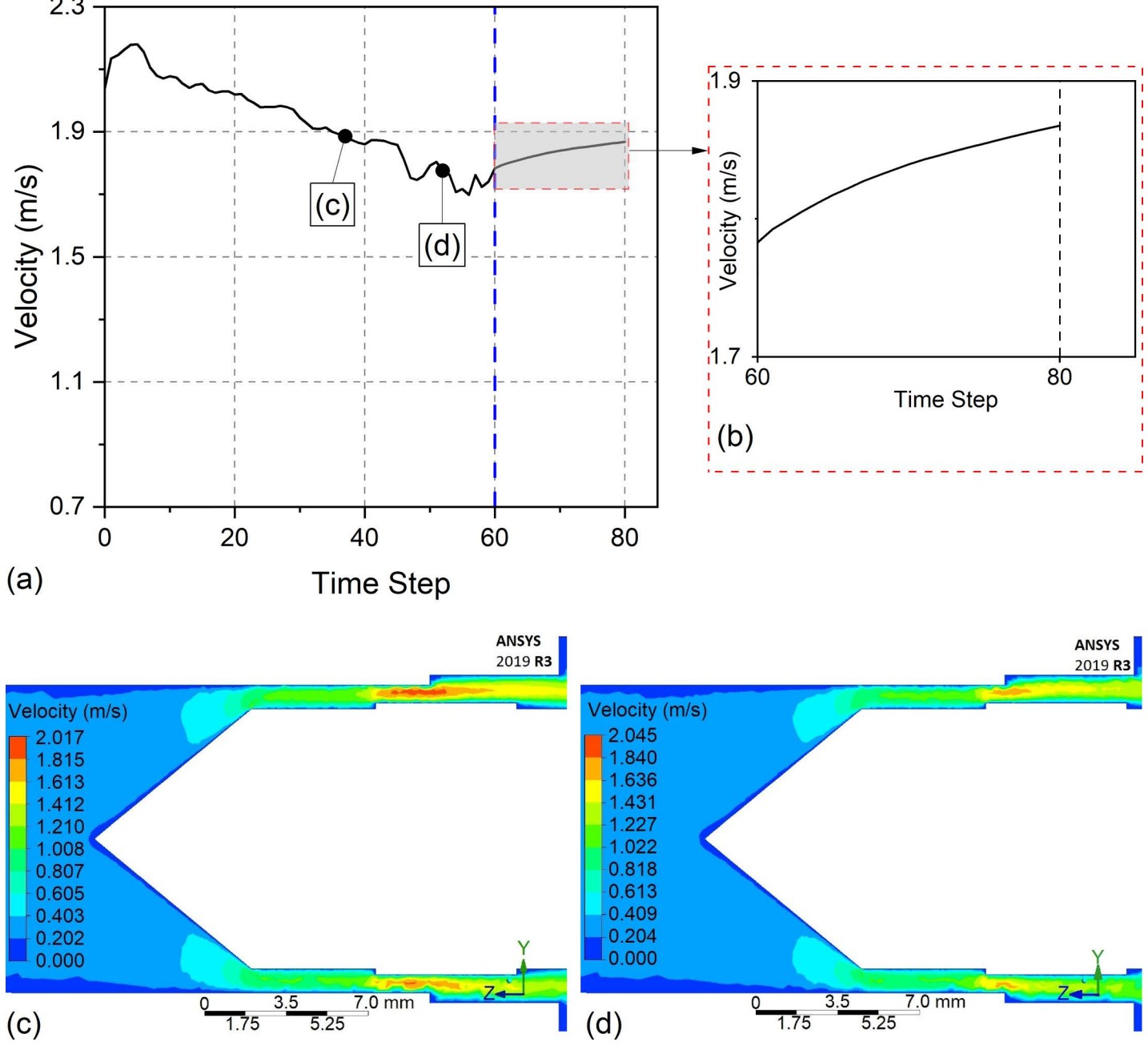

**Fig 15. Electrolyte velocity variation under the UEF condition:** (a) velocity change over time at X = 0, (b) enlarged view showing the velocity behavior when the tool begins feeding, and (c–d) electrolyte velocity distributions at the same tool positions as in Fig 14c and 14d.

on the inner walls of Stellite 21 tubes. The analysis was based on the material removal rate (MRR), current variation, and removed area (RA). The main conclusions are as follows:

- The highest MRR values of 26.67 mg/s and 24.8 mg/s were achieved during two-pass operations under HEF and UEF conditions, respectively. As the number of passes increased to 4 and 6, the MRR decreased by approximately 13% and

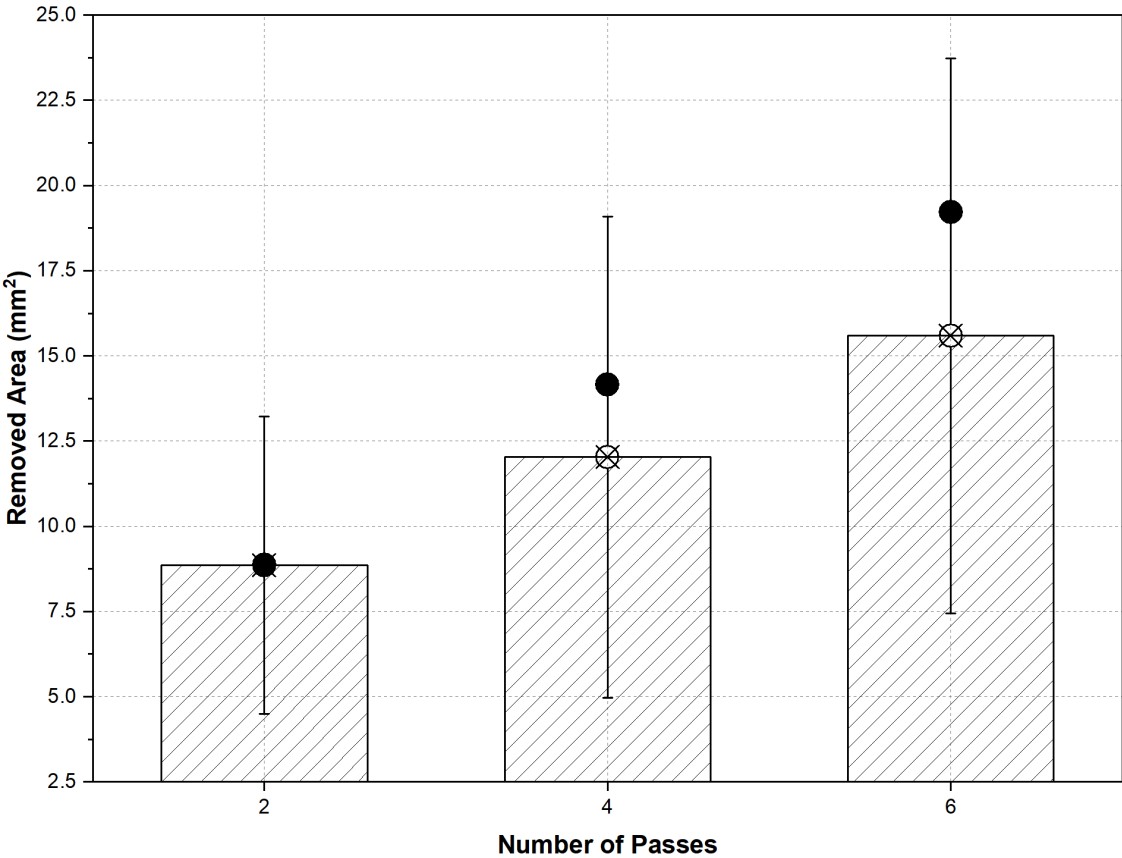

**Fig 16.  Mean of the RA (with X sign) and error lines for the different pass operations with A$_j$ (with filled ones).**

23%, respectively, accompanied by a reduction in current from 150 A to 80 A. This trend aligns with the principles of Faraday's and Ohm's laws, indicating reduced dissolution efficiency due to increased inter-electrode distance and lower current density.

• The minimum RA values of 3.9 mm² (UEF) and 6.12 mm² (HEF) were observed at the tool entry point (X = 0), which cor-responds to the lowest electrolyte velocity and conductivity as shown by simulation results. Under UEF conditions, RA increased significantly, by 293% at X = 25 and 327% at X = 50, demonstrating the process's sensitivity to electrolyte flow distribution.

• The most pronounced difference in RA was observed at X = 0 for both conditions. Specifically, the RA values at this location under HEF were 34–64% higher than those under UEF, indicating less uniform flow characteristics. Simulation results further revealed that, although velocity decreased at the X = 0 position in both conditions during tool return, the HEF condition exhibited greater fluctuations, whereas the UEF condition provided a more uniform velocity profile, result-ing in more consistent RA values.

• A comparison between the RA values calculated from the dissolved material (A$_j$) and those obtained from cross-sectional imaging confirmed the occurrence of stable machining at varying axial positions. In four- and six-pass opera-tions, the groove geometry stabilized before reaching the midpoint of the tube, highlighting the influence of accumulated flow effects on process uniformity.

These findings confirm that both electrolyte flow strategy and the number of tool passes critically affect ECM performance in internal grooving applications. The study offers practical insights into optimizing machining parameters—such as tool feed direction, pass strategy, and electrolyte distribution—to enhance geometric precision and process efficiency in industrial ECM implementations.

## Author contributions

**Funding acquisition:** Semih Ekrem Anil.

**Methodology:** Hasan Demirtas.

**Resources:** Semih Ekrem Anil.

**Writing – original draft:** Hasan Demirtas.

**Writing – review & editing:** Semih Ekrem Anil, Hasan Demirtas.

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
