## [Decision Letter · Decision Letter 0]

23 Apr 2025

Dear Dr. Demirtas,

Thank you for submitting your manuscript to PLOS ONE. After careful consideration, we feel that it has merit but does not fully meet PLOS ONE’s publication criteria as it currently stands. Therefore, we invite you to submit a revised version of the manuscript that addresses the points raised during the review process.

We look forward to receiving your revised manuscript.

Kind regards,

Julfikar Haider

Academic Editor

PLOS ONE

Journal Requirements:

“This study was funded by Savunma Sanayi Baskanligi, Presidency of the Republic of Turkiye (grant number 20SC017).”

4. We note that your Data Availability Statement is currently as follows: All relevant data are within the manuscript and in Supporting Information files.

Additional Editor Comments:

Please see the comments made by the reviewers

Reviewers' comments:

Reviewer's Responses to Questions

**Comments to the Author**

1. Is the manuscript technically sound, and do the data support the conclusions?

Reviewer #1: Yes

Reviewer #2: Partly

Reviewer #3: Partly

Reviewer #4: Yes

Reviewer #5: Yes

2. Has the statistical analysis been performed appropriately and rigorously?

Reviewer #1: N/A

Reviewer #2: No

Reviewer #3: N/A

Reviewer #4: Yes

Reviewer #5: Yes

3. Have the authors made all data underlying the findings in their manuscript fully available?

Reviewer #1: Yes

Reviewer #2: Yes

Reviewer #3: Yes

Reviewer #4: Yes

Reviewer #5: Yes

4. Is the manuscript presented in an intelligible fashion and written in standard English?

Reviewer #1: Yes

Reviewer #2: Yes

Reviewer #3: No

Reviewer #4: Yes

Reviewer #5: No

Reviewer #1: Article type: Research Article

Title: “Effects of Tool Feed Direction and Number of Passes on Electrochemical Dissolution and Groove Geometry in the Electrochemical Grooving of Tube Inner Walls”

Manus. ID: PONE-D-25-02811 has been reviewed.

- The research article is different and remarkable. However, some additions and corrections need to be made. The following list of comments will help to further improve the manuscript:

• It would be beneficial to change the title of the manuscript to a more interesting one.

• The most striking section of a study is the abstract section. Therefore, important results from the study should be highlighted.

• The following Keywords should be changed. More specific keywords should be added.

“Pressure, Unidirectional, Hybrid, Flow”

• The introduction section should be enriched with important information from the latest literature studies. (2023-2025)

• The difference (novelty) of the study from the studies in the literature should be explained more carefully. (At the end of the introduction)

• A detailed flow chart should be added to the article for a better understanding of the study.

• The “Materials and Methods” section should be rewritten in more detail.

• How the parameters applied in the study were selected should be explained in detail.

• More information should be provided about the temperature and relative humidity of the environment where the study is conducted.

• All devices/software used within the scope of the study should be given in a flow chart. (use Microsoft Visio)

• Figures 2, 4 and 8 should be rearranged and their sizes increased (3x). (single column from top to bottom)

• The results obtained within the scope of the study should be shown with at least 2-3 bar graphs. In this way, the results will be more easily understood.

• The "Unidirectional Electrolyte Flow" section should be written in more detail. (with literature support)

• There are many spelling errors in the work. Please check the section below step by step.

“….Examining the MRR values, it can be seen that although the cumulative amount of dissolved material increases, it decreases in each additional pass. This is because the change in the amount of dissolved material does not occur linearly as the machining time increases with the number of passes. The change in amperage—taking into account the initial data (150 A)— is around 20% for two passes, 40% for four passes, and 48% for six passes. The rate of decrease in the amount of current decreases as the number of passes increases. This can be attributed to how the tool is moved and as explained in the experimental work title, the tool moves in the axial direction relative to a workpiece with cylindrical geometry and does not make any movement in the radial direction. Figure 5 shows a portion of the workpiece cross-section in the radial and axial directions for passes two, four, and six. Figure 5. Cross-section of the machined part in (a) the radial direction and (b) the axial direction As seen in Figure 5, as the tool completes the number of passes, the distance between the workpiece and the tool increases, which causes the electrolyte’s electrical resistance to increase. This increase in electrical resistance limits the current flow, which causes a decrease in the amount of material machined. The amount of dissolved material for each experimental condition and the change of dissolved material are shown in Figure 6. Figure 6. Dissolved material (Δm) and change of dissolved material (ΔM) for (a) the UEF condition and (b) the HEF condition In Figure 6, is the difference in dissolved material between the two different passes, which can be calculated as follows: Δ +2 = Δ +2 − Δ , (5) where is the dissolved material and is a constant, which can be 0, 2, or 4. Although the cumulative amount of machined material constantly increased, the highest values for ΔM were obtained in the first two passes in both tool movement conditions, and the difference decreased with each subsequent pass. Relative to the first two passes, the amount of decrease was around 40% and 34% in the next four passes and 43% and 44% in the next six passes in the UEF and HEF conditions, respectively. Therefore, as the number of passes increases, the dissolution rate of the material decreases. Examining the cross-section geometry, which changes with the pass operation in the radial direction, it can be seen that the groove is made with the effect of the area where the insulation was located for the first two passes and had a more regular structure. Therefore, as the number of passes increases, the groove depth and width increase. The effect of the insulation material decreased significantly, especially after six passes, and the inner diameter of the material was enlarged (over machining). Figure 7 shows this change for the UEF condition, and the groove geometries were obtained from the middle of the workpiece. Figure 7. Changes in groove geometry for different operations As seen in Figure 6, the maximum amount of dissolved and machined area was achieved with UEF. This is because it can be related to the direction of tool movement and electrolyte transfer. With HEF, the tool moves at a constant speed, regardless of the electrolyte transfer direction. However, with UEF, the tool moves with a constant feed rate in the opposite direction of the electrolyte transfer, and during its movement in the same direction as the flow, the tool returns to its starting position very quickly (600 mm/min). Therefore, the machining operation occurs only in the opposite direction of the flow. As a result, and with the help of the conical structure of the tool’s tip, the electrolyte is forced to move between the workpiece and the tool, and the forced flow causes vortices on the inner wall of the workpiece due to the increase in pressure and speed. With HEF, half of the passes occur in the opposite direction of the flow, and the other half occur in the same direction as the flow. During its movement in the same direction as the flow, the electrolyte automatically fills the machining gap, which results in less material being machined. A schematic of this procedure is illustrated in Figure 8….”

• The “Conclusions” section should be reviewed. It should be enriched with the info/results obtained. Also, Percentage comparisons should be included. (For example... was/were 34% better. )

• • • After the revision, I would like to review the work again and see the additions and corrections made.

Reviewer #2: Dear author,

It is seen that the authors prepared an article titled "Effects of Tool Feed Direction and Number of Passes on Electrochemical Dissolution and Groove Geometry in the Electrochemical Grooving of Tube Inner Walls" and the effects of machinability characteristics on Electrochemical Dissolution and Groove Geometry were investigated. It is recommended that the manuscript be resubmitted after further investigation.

Best wishes.

Reviewer #3: 1. The abstract should be brief and concise. Please remove unnecessary explanation from the abstract.

2. Two different tool feed directions have been mentioned in the abstract. Please state them clearly. The statement is quite ambiguous.

3. The literature survey presented in the Introduction section was quite subjective. Therefore, research gaps cannot be clearly understood from the literature survey. The motivation behind this study is not clear.

4. Justify the reason for choosing the workpiece material. Please provide the physical properties of the Stellite 21.

5. In Fig. 2, which one is UEF and which one is HEF is not clear.

6. The grammar should be improved.

7. Why MRR is decreasing with an increase in number of passes? Is there any physical reason for this happening?

8. The reduction of amperage with the number of passes is not clear.

9. No significant difference has been observed between the outcomes of UEF and HEF. How can this be happened? Please give justification for showing two different flow directions.

10. What is the novelty of this study? No significant contribution has been found in this study. No clear physical justification for the presented outcomes has been found in this study.

Reviewer #4: In this paper the authors studied the impacts of different electrolyte transfer methods, unidirectional electrolyte flow (UEF) and hybrid electrolyte flow (HEF), on the EC grooving of tubes. Two parameters were considered: the number of passes and the tool feed direction. The experimental results show that although the amount of dissolved material increases with the number of passes, the incrementation rate of dissolved material decreases with the number of passes due to increased electrical resistance. Additionally, the geometry of grooves changes at the inlet and outlet of the electrolyte due to fluid properties such as pressure variance and turbulent flow. Tool feed direction also affects the uniformity of grooves.

This is a clear, concise, and well-written manuscript. The introduction is relevant, and theory based. Sufficient information about the previous study findings is presented for readers to follow the present study rationale and procedures. The text is clear and easy to read, and the results are sufficiently discussed. The objectives clearly stated, experimental methods are advanced, data statistically analyzed, the conclusions well supported by the data presented. In my opinion, the manuscript is suitable for publication as it is.

Reviewer #5: The paper does not strongly emphasize what differentiates this study from prior work. State clearly what was not done before by the investigators around the globe. Authors should the motivation for grooving inner tube walls using ECM should be expanded. What are the key challenges in conventional methods? More latest literature on recent ECM strategies for internal features or difficult-to-machine areas. Clarify the novelty, how does this study add to existing knowledge? Hardly any details related to the experimental set-up used have been discussed. Authors must add the same and explore it for better readability. Important process parameters must be clearly tabulated or presented. Was statistical analysis used to determine the significance of factors? The conclusion section is overly general. Specify key findings with numerical data. Include practical implications of the findings: How does this help optimize ECM in industry?

**Do you want your identity to be public for this peer review?** For information about this choice, including consent withdrawal, please see our Privacy Policy

Reviewer #1: No

Reviewer #2: No

Reviewer #3: No

Reviewer #4: No

Reviewer #5: No

---

## [Author Response · Author response to Decision Letter 1]

9 Jun 2025

List of Responses:

The authors acknowledge the comments of the reviewer and thank them for their comments, advice, and suggestions. All the revised parts are highlighted in yellow.

Reviewer 1

1. It would be beneficial to change the title of the manuscript to a more interesting one.

Response 1. The authors appreciate this comment and have changed the title of the manuscript.

2. The most striking section of a study is the abstract section. Therefore, important results from the study should be highlighted.

Response 2. The authors appreciate this comment and have changed the abstract section (in line 19 of page 2).

3. The following Keywords should be changed. More specific keywords should be added.

“Pressure, Unidirectional, Hybrid, Flow”

Response 3. The authors appreciate this comment and added new Keywords (Groove geometry, Stellite 21).

4. The introduction section should be enriched with important information from the latest literature studies (2023-2025).

Response 4. The authors rewrote the introduction section and added new literature (5, 6, 10, 13, 14, 21, 22, 23, 28, 29, 30) related to the presented work, and the “References” section is revised.

5. The difference (novelty) of the study from the studies in the literature should be explained more carefully. (At the end of the introduction)

Response 5. The authors thank the reviewer for this valuable comment. The authors reviewed the last paragraph of the introduction section. The revised paragraph can be found in line 103 of page 5.

6. A detailed flow chart should be added to the article for a better understanding of the study.

Response 6. The authors added a new flowchart to increase the readability of the manuscript. The new flow chart can be found in line 188 of page 9 (new Fig3). This flowchart describes how the EC grooving operation is made for different passes and feed direction strategies.

7. The “Materials and Methods” section should be rewritten in more detail.

Response 7. The authors revised the Materials and methods section, added new figures (Figs 1 and 3), and changed the old one (Figure 1) to a new one (Fig 2). Also, added info about the ECM setup (revised part can be found in line 127 of page 6), and tool geometry (revised part can be found in line 143 of page 7). The related info about the tool feed strategy and how the pass operation is made is described in detail and can be found in line 171 of page 8.

8. How the parameters applied in the study were selected should be explained in detail.

Response 8. The authors added new information in the Materials and Methods section. Also, a new table (Table 2 can be found in line 163 of page 8) is added. The parameters used in this study were selected based on the preliminary experiments by referring to [28] and can be found in line 159 of page 7.

9. More information should be provided about the temperature and relative humidity of the environment where the study is conducted.

Response 9. The authors thank the reviewer for this valuable comment. The temperature and relative humidity of the environment can be found in lines 155 to 159 of page 7. Also, these values are provided in Table 2 (in line 163 of page 7).

10. All devices/software used within the scope of the study should be given in a flow chart. (use Microsoft Visio).

Response 10. The authors added new flow charts (Figs 1 and 3) by using Microsoft Visio. These figures can be found in lines 139 of page 7 and 188 of page 8.

11. Figures 2, 4 and 8 should be rearranged and their sizes increased (3x). (single column from top to bottom).

Response 11. The authors increased the sizes of Figs 8 (old Figure 4) and 11 (old Figure 8). The increased figures can be found in lines 298 of page 14 and in line 363 of page 17. After careful consideration, and following the other reviewer's comments, the authors removed Figure 2 and added a new figure (Fig 2) and arranged its size to better explain the tool movement strategy. The related figure can be found in lines 148 of page 7.

12. The results obtained within the scope of the study should be shown with at least 2-3 bar graphs. In this way, the results will be more easily understood.

Response 12. The authors changed Fig 9 (old Figure 6), Fig 13 (old Figure 10), and Fig 16 (old Figure 12) as bar graphs to be understood by the reader more easily.

13. The "Unidirectional Electrolyte Flow" section should be written in more detail. (with literature support).

Response 13. The authors thank the reviewer for this valuable comment. The authors conducted FEM analyses to evaluate the behavior of the electrolyte flow. Therefore, the relevant section is more understandable, and the results are supported by the literature. Thus, a new section titled “Numerical simulation of electrolyte flow” (in line 210 of page 10) is added to the manuscript. Additionally, old Figure 11 is removed and new Fig 15 (in line 457 of page 21) is added to the manuscript. The “Unidirectional electrolyte flow” section is revised, and this revision can be found in lines 432 of page 20 and 463 of page 21.

14. There are many spelling errors in the work. Please check the section below step by step.

“….Examining the MRR values, it can be seen that although the cumulative amount of dissolved material increases, it decreases in each additional pass. This is because the change in the amount of dissolved material does not occur linearly as the machining time increases with the number of passes. The change in amperage—taking into account the initial data (150 A)— is around 20% for two passes, 40% for four passes, and 48% for six passes. The rate of decrease in the amount of current decreases as the number of passes increases. This can be attributed to how the tool is moved and as explained in the experimental work title, the tool moves in the axial direction relative to a workpiece with cylindrical geometry and does not make any movement in the radial direction. Figure 5 shows a portion of the workpiece cross-section in the radial and axial directions for passes two, four, and six. Figure 5. Cross-section of the machined part in (a) the radial direction and (b) the axial direction As seen in Figure 5, as the tool completes the number of passes, the distance between the workpiece and the tool increases, which causes the electrolyte’s electrical resistance to increase. This increase in electrical resistance limits the current flow, which causes a decrease in the amount of material machined. The amount of dissolved material for each experimental condition and the change of dissolved material are shown in Figure 6. Figure 6. Dissolved material (Δm) and change of dissolved material (ΔM) for (a) the UEF condition and (b) the HEF condition In Figure 6, is the difference in dissolved material between the two different passes, which can be calculated as follows: Δ +2 = Δ +2 − Δ , (5) where is the dissolved material and is a constant, which can be 0, 2, or 4. Although the cumulative amount of machined material constantly increased, the highest values for ΔM were obtained in the first two passes in both tool movement conditions, and the difference decreased with each subsequent pass. Relative to the first two passes, the amount of decrease was around 40% and 34% in the next four passes and 43% and 44% in the next six passes in the UEF and HEF conditions, respectively. Therefore, as the number of passes increases, the dissolution rate of the material decreases. Examining the cross-section geometry, which changes with the pass operation in the radial direction, it can be seen that the groove is made with the effect of the area where the insulation was located for the first two passes and had a more regular structure. Therefore, as the number of passes increases, the groove depth and width increase. The effect of the insulation material decreased significantly, especially after six passes, and the inner diameter of the material was enlarged (over machining). Figure 7 shows this change for the UEF condition, and the groove geometries were obtained from the middle of the workpiece. Figure 7. Changes in groove geometry for different operations As seen in Figure 6, the maximum amount of dissolved and machined area was achieved with UEF. This is because it can be related to the direction of tool movement and electrolyte transfer. With HEF, the tool moves at a constant speed, regardless of the electrolyte transfer direction. However, with UEF, the tool moves with a constant feed rate in the opposite direction of the electrolyte transfer, and during its movement in the same direction as the flow, the tool returns to its starting position very quickly (600 mm/min). Therefore, the machining operation occurs only in the opposite direction of the flow. As a result, and with the help of the conical structure of the tool’s tip, the electrolyte is forced to move between the workpiece and the tool, and the forced flow causes vortices on the inner wall of the workpiece due to the increase in pressure and speed. With HEF, half of the passes occur in the opposite direction of the flow, and the other half occur in the same direction as the flow. During its movement in the same direction as the flow, the electrolyte automatically fills the machining gap, which results in less material being machined. A schematic of this procedure is illustrated in Figure 8….”

Response 14. The authors checked all spelling errors and changed the dedicated sentences. Also, the manuscript is checked for grammar errors and revised by the authors.

15. The “Conclusions” section should be reviewed. It should be enriched with the info/results obtained. Also, Percentage comparisons should be included. (For example... was/were 34% better.).

Response 15. The authors appreciate to reviewer for this valuable comment. We believe that this revision makes this manuscript more valuable. Therefore, the authors changed the “Conclusion” section, and the relevant revision can be found in line 477 of page 22.

• • • After the revision, I would like to review the work again and see the additions and corrections made.

• • • Thanks for the valuable comments and suggestions to make this manuscript better.

NOTE: Equation and figure numbering are listed according to the revised manuscript. The revised section in the main text body is colored according to the response colors shown in this file. The page and line numbers mentioned herein are based on the manuscript version submitted without figures.

The authors acknowledge the comments of the reviewer and thank them for their comments, advice, and suggestions.

Reviewer 2

• Dear author,

It is seen that the authors prepared an article titled "Effects of Tool Feed Direction and Number of Passes on Electrochemical Dissolution and Groove Geometry in the Electrochemical Grooving of Tube Inner Walls" and the effects of machinability characteristics on Electrochemical Dissolution and Groove Geometry were investigated. It is recommended that the manuscript be resubmitted after further investigation.

Best wishes.

Response to reviewer:

Dear Reviewer,

We sincerely thank you for your valuable comments and suggestions regarding our manuscript.

In response to your feedback, we have made significant improvements to the manuscript. The content has been enriched by adding new sections, including detailed numerical flow analyses to better explain the electrolyte behavior, as well as updated and clearer figures to illustrate our findings more effectively. Furthermore, the introduction and discussion sections have been expanded to provide a deeper understanding of the experimental results and their implications.

Additionally, the title and abstract have been revised to better reflect the scope and key outcomes of our study. We believe these revisions substantially improve the quality and clarity of the manuscript.

We appreciate your thoughtful evaluation, which has helped us enhance the manuscript considerably. We kindly invite you to review the revised version and look forward to your constructive feedback.

Thank you once again for your time and valuable insights.

Best regards,

Hasan

NOTE: Equation and figure numbering are listed according to the revised manuscript. The revised section in the main text body is colored according to the response colors shown in this file. The page and line numbers mentioned herein are based on the manuscript version submitted without figures.

The authors acknowledge the comments of the reviewer and thank them for their comments, advice, and suggestions.

Reviewer 3

1. The abstract should be brief and concise. Please remove unnecessary explanation from the abstract.

Response 1. We appreciate this valuable comment and have revised the abstract to make it more brief and concise by removing unnecessary explanations (in line 19 of page 2).

2. Two different tool feed directions have been mentioned in the abstract. Please state them clearly. The statement is quite ambiguous.

Response 2. Thank you for pointing this out. We have clearly defined the two tool feed directions in the revised abstract to remove any ambiguity. The updated abstract is highlighted in yellow and can be found in line 19 of page 2.

3. The literature survey presented in the Introduction section was quite subjective. Therefore, research gaps cannot be clearly understood from the literature survey. The motivation behind this study is not clear.

Response 3. Based on the reviewer’s and other reviewers’ feedback, we rewrote the Introduction section to provide a more objective and comprehensive literature review. Additionally, we explicitly clarified the research gaps and motivation behind this study. These revisions start on page 3, line 69, highlighted in yellow.

4. Justify the reason for choosing the workpiece material. Please provide the physical properties of the Stellite 21.

Response 4. We thank the reviewer for this suggestion. The reasons for selecting Stellite 21 as the workpiece material have been added to the manuscript (page 3, line 43). Furthermore, the physical and mechanical properties of Stellite 21 are now presented in Table 1 (page 6, line 126), with all additions highlighted in green.

5. In Fig. 2, which one is UEF and which one is HEF is not clear.

Response 5. After careful consideration and in response to the comments of other reviewers, we replaced the old Figure 2 with a new figure (Fig 2) that clearly distinguishes between UEF and HEF conditions (page 7, line 149). Detailed explanations of UEF and HEF are also included in the “Materials and methods” section, complemented by a newly added flow chart (Fig 3). Relevant text is highlighted in yellow and can be found in line 171 of page 8.

6. The grammar should be improved.

Response 6. We appreciate the reviewer’s comment and have thoroughly revised the manuscript for improved grammar and language clarity throughout the text.

7. Why MRR is decreasing with an increase in number of passes? Is there any physical reason for this happening?

Response 7. Additional explanations regarding the physical reasons behind the observed decrease in material removal rate (MRR) as the number of passes increases have been incorporated. Please refer to page 15, line 315, highlighted in green.

8. The reduction of amperage with the number of passes is not clear.

Response 8. We added a detailed discussion on the relationship between amperage and the number of passes to clarify this phenomenon, located on page 15, line 309, also highlighted in green.

9. No significant difference has been observed between the outcomes of UEF and HEF. How can this be happened? Please give justification for showing two different flow directions.

Response 9. Thank you for this insightful question. Although initial results showed minimal differences, we observed the most notable variations at the tool entry point (X=0), as demonstrated in Fig 12. To investigate this further, we performed finite element method (FEM) simulations of electrolyte flow, and added a new section titled “Numerical simulation of electrolyte flow” (page 10, line 210). Based on these analyses, we revised the “Unidirectional electrolyte flow” (page 20, line 429) and “Hybrid electrolyte flow” (page 18, line 383) sections, providing a detailed discussion supported by literature. Old Figure 11 was removed

---

## [Decision Letter · Decision Letter 1]

3 Aug 2025

Dear Dr. Demirtas,

We look forward to receiving your revised manuscript.

Kind regards,

Mithilesh K. Dikshit

Academic Editor

PLOS ONE

Journal Requirements:

Reviewers' comments:

Reviewer's Responses to Questions

**Comments to the Author**

Reviewer #1: All comments have been addressed

Reviewer #2: All comments have been addressed

Reviewer #3: (No Response)

Reviewer #4: All comments have been addressed

Reviewer #5: All comments have been addressed

2. Is the manuscript technically sound, and do the data support the conclusions?

Reviewer #1: Yes

Reviewer #2: Yes

Reviewer #3: Partly

Reviewer #4: Yes

Reviewer #5: Yes

3. Has the statistical analysis been performed appropriately and rigorously?

Reviewer #1: Yes

Reviewer #2: No

Reviewer #3: N/A

Reviewer #4: Yes

Reviewer #5: Yes

4. Have the authors made all data underlying the findings in their manuscript fully available?

Reviewer #1: Yes

Reviewer #2: Yes

Reviewer #3: Yes

Reviewer #4: Yes

Reviewer #5: Yes

5. Is the manuscript presented in an intelligible fashion and written in standard English?

Reviewer #1: Yes

Reviewer #2: Yes

Reviewer #3: Yes

Reviewer #4: Yes

Reviewer #5: (No Response)

Reviewer #1: After revision:

The authors have made significant improvements to the paper by addressing the feedback provided by the reviewers, resulting in a clearer presentation of results. Based on these revisions, the paper is now ready for acceptance.

Reviewer #2: Dear author,

Necessary revisions have been performed. Manuscript can be accepted for publishing.

Best wishes.

Reviewer #3: 1. The abstract is too much extended. It is better to shorten the abstract and display only the key findings.

2. The literature survey not only includes the description of the work defined in the previous literature. However, it is the combination of the summary and the critics of each literature. The current literature survey lacks that point. The author should take care of it while writing the introduction.

3. Fig. 2 does not show any UEF and HEF conditions.

4. The initial results showed minimal differences between the outcomes achieved from UEF and HEF conditions. However, the current results show significant differences between the outcomes from UEF and HEF, especially in the entry position as explained in the revised manuscript. How many times have the experiments been repeated? If the results are repeatable, why was the difference not significant in the initial study?

Reviewer #4: The authors have answered all the questions and improved the paper. The paper can be published in its present form.

Reviewer #5: Authors have now incorporated all the suggestions. The article is now well revised and improved as well.

**Do you want your identity to be public for this peer review?** For information about this choice, including consent withdrawal, please see our Privacy Policy

Reviewer #1: No

Reviewer #2: No

Reviewer #3: No

Reviewer #4: **Yes: ** Siamak Ghorbani

Reviewer #5: No

---

## [Author Response · Author response to Decision Letter 2]

19 Aug 2025

List of Responses:

The authors acknowledge the comments of the reviewer and thank them for their comments, advice, and suggestions.

Reviewer 1

The authors have made significant improvements to the paper by addressing the feedback provided by the reviewers, resulting in a clearer presentation of results. Based on these revisions, the paper is now ready for acceptance.

Response to Reviewer 1:

We sincerely thank the reviewer for the encouraging and positive feedback. We greatly appreciate your recognition of the clarity, relevance, and scientific rigor of our study. Your supportive comments are truly motivating and have reinforced the value of our work.

Thank you again for your time and kind evaluation.

The authors acknowledge the comments of the reviewer and thank them for their comments, advice, and suggestions.

Reviewer 2: Dear author,

Necessary revisions have been performed. Manuscript can be accepted for publishing.

Best wishes.

Response to Reviewer 2:

We sincerely thank the reviewer for the encouraging and positive feedback. We greatly appreciate your recognition of the clarity, relevance, and scientific rigor of our study. Your supportive comments are truly motivating and have reinforced the value of our work.

Thank you again for your time and kind evaluation.

Reviewer 3

The authors acknowledge the comments of the reviewer and thank them for their comments, advice, and suggestions. All the revised parts are highlighted in yellow.

1. The abstract is too much extended. It is better to shorten the abstract and display only the key findings.

Response 1. We appreciate this valuable comment and have revised the abstract to make it more brief and concise by removing unnecessary explanations (in line 20 of page 2).

2. The literature survey not only includes the description of the work defined in the previous literature. However, it is the combination of the summary and the critics of each literature. The current literature survey lacks that point. The author should take care of it while writing the introduction.

Response 2. We thank the reviewer for the valuable suggestion regarding the literature survey. In the revised manuscript, we have carefully addressed this concern by not only summarizing the key findings of previous studies but also critically evaluating their strengths and limitations. Specifically, for each cited work, we have added discussion on the parameters or conditions that were not considered, such as electrolyte flow rate, tool feed rate, temperature distribution, and boundary conditions, and we have highlighted how these limitations motivated the present study. These changes have been incorporated throughout the introduction section to ensure a comprehensive and critical literature review. (starting from line 64 of page 3).

3. Fig. 2 does not show any UEF and HEF conditions.

Response 3. Thank you for your valuable comment. Accordingly, illustrations for the HEF and UEF conditions have been added as Fig. 2d and Fig. 2e, respectively. In-text citations referring to these subfigures have also been included to improve clarity.

4. The initial results showed minimal differences between the outcomes achieved from UEF and HEF conditions. However, the current results show significant differences between the outcomes from UEF and HEF, especially in the entry position as explained in the revised manuscript. How many times have the experiments been repeated? If the results are repeatable, why was the difference not significant in the initial study?

Response 4. The experimental results presented in the revised manuscript are consistent with those reported in the original submission. In the 2nd revision, we have included additional analyses and simulations to provide a clearer explanation of the differences observed between UEF and HEF conditions, particularly at the entry position. The experiments were repeated three times to ensure repeatability, and the outcomes showed high consistency with the initial observations. Since the variations across repetitions were negligible, error bars were not included in the figures. The relevant information can be found in the original manuscript (see page 12, under “Analysis of the Machined Area” section; page 15, under “Unidirectional Electrolyte Flow” section; page 18, under “Conclusions” section in the eighth conclusion). The enhanced analyses, therefore, aim to clarify why significant differences appear in specific regions, without altering the originally reported data.

NOTE: Equation and figure numbering is listed according to the revised manuscript. The revised section in the main text body are colored according to the response colors shown in this file. The page and line numbers mentioned herein are based on the manuscript version submitted without figures.

The authors acknowledge the comments of the reviewer and thank them for their comments, advice, and suggestions.

Reviewer 4

The authors have answered all the questions and improved the paper. The paper can be published in its present form.

Response to reviewer:

We sincerely thank the reviewer for the encouraging and positive feedback. We greatly appreciate your recognition of the clarity, relevance, and scientific rigor of our study. Your supportive comments are truly motivating and have reinforced the value of our work.

Thank you again for your time and kind evaluation.

The authors acknowledge the comments of the reviewer and thank them for their comments, advice, and suggestions.

Reviewer 5

Authors have now incorporated all the suggestions. The article is now well revised and improved as well.

Response to reviewer:

We sincerely thank the reviewer for the encouraging and positive feedback. We greatly appreciate your recognition of the clarity, relevance, and scientific rigor of our study. Your supportive comments are truly motivating and have reinforced the value of our work.

Thank you again for your time and kind evaluation.

NOTE: Equation and figure numbering are listed according to the revised manuscript. The revised section in the main text body is colored according to the response colors shown in this file. The page and line numbers mentioned herein are based on the manuscript version submitted without figures.

---

## [Decision Letter · Decision Letter 2]

24 Aug 2025

Electrochemical grooving of tube inner walls with emphasis on feed strategy and multi-pass effects on material removal and groove geometry

PONE-D-25-02811R2

Dear Dr. Demirtas,

We’re pleased to inform you that your manuscript has been judged scientifically suitable for publication and will be formally accepted for publication once it meets all outstanding technical requirements.

Kind regards,

Dr. Mithilesh K. Dikshit

Academic Editor

PLOS ONE

Additional Editor Comments (optional):

The manuscript is accepted in the current format.

Reviewers' comments:

Reviewer's Responses to Questions

**Comments to the Author**

Reviewer #3: All comments have been addressed

2. Is the manuscript technically sound, and do the data support the conclusions?

Reviewer #3: Yes

3. Has the statistical analysis been performed appropriately and rigorously?

Reviewer #3: N/A

4. Have the authors made all data underlying the findings in their manuscript fully available?

Reviewer #3: Yes

5. Is the manuscript presented in an intelligible fashion and written in standard English?

Reviewer #3: Yes

Reviewer #3: All the comments have been addressed appropriately. This paper can be accepted for publication in the present form.

**Do you want your identity to be public for this peer review?** For information about this choice, including consent withdrawal, please see our Privacy Policy

Reviewer #3: No

---

## [Editor Report · Acceptance letter]

PONE-D-25-02811R2

PLOS ONE

Dear Dr. Demirtas,

I'm pleased to inform you that your manuscript has been deemed suitable for publication in PLOS ONE. Congratulations! Your manuscript is now being handed over to our production team.

Kind regards,

on behalf of

Dr. Mithilesh K. Dikshit

Academic Editor

PLOS ONE